# Networked Digital Public Goods Games with Heterogeneous Players and Convex Costs

## ABSTRACT

In the digital age, resources such as open-source software and publicly accessible databases form a crucial category of digital public goods, providing extensive benefits across the Internet. However, the inherent non-exclusivity and non-competitiveness of these public goods frequently result in under-provision, a dilemma exacerbated by individuals' tendency to free-ride. This scenario fosters both cooperation and competition among users, leading to the emergence of public goods games.

This paper investigates networked public goods games involving heterogeneous players and convex costs to explore solutions of Nash Equilibrium (NE) for this problem. In these games, each player can choose her own effort level, representing the contributions to public goods. We employ network structures to depict the interactions among participants. Each player's utility is composed of a *concave* value component, influenced by collective efforts, and a *convex* cost component, determined solely by individual effort. To the best of our knowledge, this study is the first to explore a networked public goods game with convex costs.

Our research begins by examining welfare solutions aimed at maximizing social welfare and ensuring the convergence of pseudo-gradient ascent dynamics. We establish the presence of NE in this model and provide an in-depth analysis of the conditions under which NE is unique. Additionally, we introduce the concept of game equivalence, which expands the range of public goods games that can support a unique NE. We also delve into *comparative statics*, an essential tool in economics, to evaluate how slight modifications in the model—interpreted as monetary redistribution—impact player utilities. In addition, we analyze a particular scenario with a predefined game structure, illustrating the practical relevance of our theoretical insights. Consequently, our research enhances the broader understanding of strategic interactions and structural dynamics in networked public goods games, with significant implications for policy design in internet economic and social networks.

## CCS CONCEPTS

• **Do Not Use This Code → Generate the Correct Terms for Your Paper**; *Generate the Correct Terms for Your Paper*; Generate the Correct Terms for Your Paper; Generate the Correct Terms for Your Paper.

## KEYWORDS

Public Good Games, Networks, Nash Equilibrium, Social Welfare, Pseudo-Gradient Dynamics, Comparative Statics.

**ACM Reference Format:**
Anonymous Author(s). 2018. Networked Digital Public Goods Games with Heterogeneous Players and Convex Costs. In *Proceedings of Make sure to enter the correct conference title from your rights confirmation emai (Conference WebConf '25)*. ACM, New York, NY, USA, 20 pages. https://doi.org/XXXXXXX.XXXXXXX

## 1 INTRODUCTION

The concept of public goods is not only a significant area of interest in economic research but also closely related to the web era. Public goods encompass a wide range of web resources, in the forms like open-source software (*e.g.*, GitHub), public databases (*e.g.*, MNIST), scientific technologies (*e.g.*, papers in The WebConf), and widely accessible scientific knowledge (*e.g.*, Wikipedia, Stack overflow). The defining characteristics of public goods are their non-excludability, meaning all community members can freely use these resources without excluding any-one, and non-rivalry, where one person's use does not diminish the availability for others. Such characteristics are particularly notable in the internet. Web and internet research delves into how to effectively provide and manage digital public goods to maximize social welfare. This exploration is not just theoretical but also has practical implications for policy and development of website content, attracting an increasing number of researchers to this burgeoning field.

However, from a societal perspective, digital public goods often face challenges due to insufficient provision, a problem frequently attributed to the issue of free-riding. Consequently, each participant must decide how much effort to contribute when investing in digital public goods, aware that their efforts will also benefit others. This strategic decision-making process embodies what is known as a *public goods game*. This game can reveal complex interactions between cooperation and competition, as individuals shall balance their personal contributions against the collective benefits. Much of the prior research [Bramoullé and Kranton 2007] has focused on idealized models where participants are assumed to be homogeneous. However, in reality, especially in the case of digital public goods, users exhibit significant heterogeneity. For example, a specialized dictionary on Wikipedia is more beneficial to those within the relevant field. On the other hand, in the context of paper reviews, the efforts of one reviewer benefit the entire conference but may disadvantage the author of a low-quality submission. This demonstrates that the impact of a public good (or bad) can be either positive or negative, and varies across different participants.

In this paper, we are more interested in the networked public goods games, which effectively capture the social connections between individuals. Specifically, all participants are positioned at

the vertices of the network, and the links—each weighted differently—represent the relationships and influence between any two participants [Li et al. 2023]. Bramoullé and Kranton [2007] pioneered the study of public goods game within a network. In their homogeneous model, the utility functions of all players are consistently formulated as $u_i(x) = f(x_i + \sum_{j \in N_i} x_j) - cx_i$, where $x_i$ is the effort level of player $i$, $f(\cdot)$ is a homogeneous benefit function applicable to all players, and the cost function is linear, characterized by a uniform unit cost $c$ for all players. Furthermore, this model is unweighted, as each player exhibits the same preference for both their own efforts and those of others when computing the benefit. Based on this simplified and idealistic setting, Bramoullé and Kranton [2007] demonstrated the existence of an equilibrium where some players exert the same maximum effort while all others engage in free riding. Moreover, they showed that those contributing positive effort form an independent set within the network. While later studies have explored the public goods games with heterogeneous utility functions [Bayer et al. 2023; Papadimitriou and Peng 2021], their focus remained on the linear cost scenarios.

However, practical scenarios often feature non-linear cost functions, particularly evident in digital public goods. For instance, the initial setup of a Wikipedia article involves adding basic facts and general information—tasks that are relatively low in cost. Yet, as the article develops, ensuring accuracy and providing in-depth analysis demand increasingly specialized knowledge, research, and citations, raising the marginal cost of contributions. Unfortunately, the predominant body of research on public goods games focuses on linear cost functions [Bayer et al. 2023; Bramoullé and Kranton 2007; López-Pintado 2013; Papadimitriou and Peng 2021], and very few studies delve into the implications of non-linear cost functions.

This paper presents a novel model of networked public goods games that incorporates convex cost functions, aiming at understanding the equilibrium and dynamic in the field of digital public goods. Specifically, given an effort profile $\mathbf{x} = (x_1, x_2, \cdots, x_n)$, each player's payoff is determined by the net gain, which is the difference between a benefit function $f_i(k_i)$ and a cost function $c_i(x_i)$. The benefit function $f_i$ for player $i$ is both concave and strictly increasing, and it is derived from the gain $k_i$. This gain $k_i$ is computed as a weighted linear combination of the efforts of both the player and her neighbors. The cost function $c_i$, which is convex and strictly increases, depends exclusively on the player's own effort $x_i$.

### 1.1 Results and Techniques

Our work is the first one to study the networked public good games with convex costs. The heterogeneity of benefit functions and cost functions lends greater generality to the networked pubic goods game studied in this paper. We start at exploring the concept of welfare solutions, focusing on the maximization of social welfare and the investigation of pseudo-gradient ascent dynamics, which shows insight on the following analysis. We carefully analyze the existence and uniqueness of Nash Equilibrium (NE) across various settings, providing deep insights into the NE's structures in public goods games. Our examination extends to cases in which distinct characteristics of cost and benefit functions play a crucial role in ensuring the NE's uniqueness. Building on these foundations, we delve into comparative statics to assess the effects of subtle shifts in

the model's parameters, which we regard as money redistribution, on the utilities of the players involved. Comparative statics is a crucial analytical method in economics. This element of our study illuminates how minor adjustments can significantly influence economic outcomes and player behaviors within the game. We also studies a special case, in which the game structure is pre-defined and show how these theorems can be applied into this case.

The proof of the existence of a Nash Equilibrium (NE) primarily relies on the application of the Brouwer fixed-point theorem. Brouwer fixed-point theorem states that any continuous function mapping a compact, convex set to itself must have a fixed point [Brouwer 1911]. It's important to note that the best-response function is continuous when the utility functions are strictly concave. The proof then proceeds through a strategic modification of the utility functions, ensuring they meet the criteria stipulated by Brouwer fixed-point theorem.

To carve out the uniqueness of NE, we bring out the concept of near-potential game, and show that under certain condition, the NE of near-potential game is unique, and pseudo-gradient ascent dynamic will converges to this point with exponential rate. The proof constructs the discrete version of pseudo-gradient ascent dynamic and show that it is compressive mapping, which is guaranteed to have unique fixed point by Banach's theorem [Banach 1922]. We then bridge the gap between near-potential game and public good games, showing three conditions under which we can transform the public good games into a specifically-designed near-potential game while holding the NEs invariant, therefore guarantee the uniqueness of NE. We also proposes the concept of game equivalence, that ensures the one-to-one relationships between the NEs of corresponding games, which can also broaden the class of games possessing unique NE.

To study the comparative statics on money redistribution, we mainly use the high-dimensional implicit function theorem. We rewrite the conditions of Nash equilibrium $x^*$ as an implicit function of infinitesimal model change $t\delta$ and corresponding NE $x^*(t)$. By differentiating this implicit function on $t$, we can derive the relation between $x^*(t)$, $\delta$ and $t$.

### 1.2 Related Works

*Public Goods in the Web Era.* In the web era, public goods play a crucial role in fostering collaborative contributions and maintaining online platforms. Gallus [2017] demonstrates the impact of symbolic awards on volunteer retention in a public goods setting like Wikipedia, where recognition and community engagement can encourage sustained contributions without direct financial incentives. Similarly, the challenges of knowledge-sharing in Web 2.0 communities have been framed as a public goods problem, where social dilemmas like free-riding are mitigated through enhanced group identity and pro-social behavior [Allen 2010]. Experimental research on cooperation in web-based public goods games further examines how network structures influence contribution behavior, with findings suggesting that contagion effects in cooperative behavior are limited to direct network neighbors [Suri and Watts 2011]. Moreover, the broader economic dynamics of the web are analyzed through the concept of "web goods," where users contribute content, exchange information, and interact in a socio-economic system that

requires balancing open access with incentive structures for content production and infrastructure development [Vafopoulos et al. 2012]. These works collectively highlight the unique challenges and opportunities of managing public goods in the digital age, emphasizing the importance of community-driven incentives and network effects in fostering web-based cooperation.

*Networked Public Good Games.* Bramoullé and Kranton [2007] initiated the study of public goods in a network. They studied the public good games on an unweighted, undirected networks with linear cost functions and homogeneous players. Under their models, there is an unique level $k^*$ such that it's optimal for each player to make the sum efforts within her neighbourhoods to be $k^*$, which greatly simplifies the analysis of the model. The authors showed that the NE of the game corresponds to the maximal independent set, where the player in the maximal independent set asserts full effort $k^*$, and the players outside free-ride.

There are many other works following this literature, see [Allouch 2015; Boncinelli and Pin 2012; Bramoullé et al. 2014; Elliott and Golub 2021]. Bramoullé et al. [2014] extended the model to the imperfectly substitute public goods case, and proved the existence and uniqueness of Nash equilibrium, under the condition of sufficiently small lowest eigenvalue of the graph matrix. Allouch [2015] differentiated the provision of public goods and private goods, and their results of existence and uniqueness of Nash equilibrium also relies on the lowest eigenvalue of the graph matrix. López-Pintado [2013] began with the studies of public good games in directed networks, by discussing both of the static model and the dynamic model. To be specific, in the static model, all players are situated within a fixed network where they choose their actions simultaneously. López-Pintado [2013] demonstrated that the structure of Nash equilibria correlates with the maximal independent set. In contrast, the dynamic model is characterized by a dynamic sampling process, where agents periodically sample a subset of other agents and base their decisions on a myopic-best response. The author established the existence of a unique globally stable proportion of public good providers in this model. Bayer et al. [2023] studied the convergence of best response dynamic on the public good games in directed networks.

A significant networked public goods game variant considers indivisible goods, where players can only make binary decisions[Galeotti et al. 2010]. Building upon this binary networked public goods (BNPG) game model, Yu et al. [2020] introduced the algorithmic inquiry of determining the existence of pure-strategy Nash equilibrium (PSNE). Specifically, they investigated the existence of PSNE in the BNPG game and proved that it is NP-hard in both homogeneous and heterogeneous settings. The computational complexity of public goods games with a network structure, such as tree or clique [Maiti and Dey 2024; Yang and Wang 2020], and regular graph [Feldman et al. 2013] has also been extensively studied. Papadimitriou and Peng[Papadimitriou and Peng 2021] proved that finding an approximate NE of the public good games in directed networks is PPAD-hard, even the utility is in a summation form. Subsequently, Gilboa and Nisan [2022] modeled players as different patterns and showed that the existence of PSNE on some non-trivial patterns is NP-complete, while a polynomial time algorithm exists for some specific patterns. In addition, Klimm and Stahlberg

[2023] further demonstrated the complexity results of the BNPG game on undirected graphs with different utility patterns to be NP-hard. They also showed that computing equilibrium in games with integer weight edges is PLS-complete.

*Continuous-time Public Good Games.* A branch of the literature on public goods focuses on studying the dynamic provision of public goods in continuous time. Fershtman and Nitzan [1991] were the first to explore this problem. They proposed two equilibrium concepts: the open-loop equilibrium and the feedback equilibrium, showing that in the feedback equilibrium, the players' utilities are lower than in the open-loop equilibrium. This result is derived under the linear strategy assumption of the feedback equilibrium, as the feedback equilibrium is not generally unique. Later, Wirl [1996] discovered that if non-linear strategies are allowed in the feedback equilibrium, it is possible for players' utilities to be higher in some feedback equilibria than in the open-loop equilibrium. Fujiwara and Matsueda [2009] generalized these findings to more general utility functions and confirmed that the results still hold. Wang and Ewald [2010] extended this work by considering environments with uncertainty.

Although these studies present findings in dynamic scenarios, they generally assume homogeneity among players in terms of utility functions (both gains and costs) and interpersonal relationships, and thus do not take network effects into consideration. To the best of our knowledge, there has been no previous research that simultaneously explores the dynamic provision of public goods with heterogeneous players.

*Concave Games.* Rosen [1965] firstly introduced the concept of concave games, in which the utility function of each player is concave with respect to her own strategy. In this paper, Rosen [1965] provided a sufficient condition for such games to have a unique equilibrium and introduced a differential equation that converges to this equilibrium. Because of the foundational results of Rosen [1965], several works have extended the study of concave games in various settings, such as learning perspective of equilibrium in concave games [Bravo et al. 2018; Mertikopoulos and Zhou 2019; Nesterov 2009], equilibrium concept in concave games [Forgó 1994; Goktas and Greenwald 2021; Ui 2008]. However, there is limited research applying the concave games framework to public goods scenarios. Our work is pioneering in applying the convex game framework to public goods games. We demonstrate that public goods games can be treated as a specific type of concave game, called near-potential game, where the potential function is meticulously designed for diverse scenarios. The uniqueness of equilibrium in near-potential games therefore directly supports the uniqueness of equilibrium in public goods games.

## 2 MODELS

Consider a community with $n$ players playing a public good game. Each player $i$ needs to decide her effort $x_i \in [\underline{x}_i, \bar{x}_i] \coloneqq X_i$ to invest the public goods, where $\{\underline{x}_i, \bar{x}_i\}_{i \in [n]}$ are predetermined and public known. Let $\boldsymbol{x} = (x_1, ..., x_n)$ be the effort profile of all players, and $\boldsymbol{x}_{-i}$ be the effort profile of all players without player $i$. Therefore, $(y_i, \boldsymbol{x}_{-i})$ is the effort profile that player $i$ chooses $y_i$ and other players keep their choices the same as $\boldsymbol{x}_{-i}$. Similarly, define $X =$

$\times_{i \in [n]} X_i$ and $X_{-i} = \times_{j \neq i} X_j$. Let us denote $W = \{w_{ij}\}_{i,j \in [n]}$ as the matrix, in which $w_{ij}$ represents the marginal gain of player $i$ from player $j$'s effort. We normalize $W$ such that $w_{ii} = 1$ for any $i \in [n]$ without loss of generality. Our model is more general as it imposes no additional constraints on the network, e.g., $w_{ij} \in [0, 1]$ or $w_{ij} = w_{ji}$, and any $w_{ij} \in \mathbb{R}$ are permitted, provided that $w_{ij} = 0$ if there is no edge between $i$ and $j$. Let $k_i \in K_i$ be the total gain of player $i$ and $\boldsymbol{k} = (k_1, ..., k_n) \in K$ be the gain profile of all players. Then we have $k_i = \sum_{j \in [n]} w_{ij} x_j$, i.e., the gain of player $i$ linearly depends on her own and other players' efforts, weighted by $\boldsymbol{w}_i = (w_{ij})_{j \in [n]}$. Therefore, $\boldsymbol{k} = W\boldsymbol{x}$. In addition, we assume $K_i = [\underline{k}_i, \bar{k}_i]$, where $\underline{k}_i$ and $\bar{k}_i$ are the minimum and maximum possible gain for player $i$ for ease of representation, respectively[1]. Similarly we use $K = \times_{i \in [n]} K_i$ and define $K_{-i} = \times_{j \neq i} K_j$.

Given an effort profile $\boldsymbol{x} = (x_1, x_2, \cdots, x_n)$, each player $i$ has utility function $u_i(\boldsymbol{x}) = f_i(k_i) - c_i(x_i)$, where $f_i$ is a concave and strictly increasing function on $K_i$, and $c_i$ is a convex and strictly increasing function on $X_i$. That is, $f_i''(x) \leq 0, f_i'(x) > 0, c_i'(x) > 0, c_i''(x) \geq 0$, meaning that $f_i$ and $c_i$ are twice differentiable. Thus, a networked public goods game $G$ is represented by a four-tuple, $G = \langle \{f_i\}_{i \in [n]}, \{c_i\}_{i \in [n]}, \{X_i\}_{i \in [n]}, W \rangle$, where $K_i$ is omitted since it can be uniquely determined from $G$.

Utility function $u_i(\boldsymbol{x}) = f_i(k_i) - c_i(x_i)$ indicates that each player's utility is composed of two parts, the value part $f_i(k_i)$ and the cost part $c_i(x_i)$. Clearly, the value part depends on her gains $k_i$, which are positively or negatively effected from other players' efforts and the cost part only depends on her own effort. Notice that one's effort will increase or decrease others' gains, and so will their utilities. Therefore players' efforts can be regarded as public goods (bads).

For the sake of convenience, we use $\mathbb{R}_+$ to denote the set of non-negative real numbers and $\mathbb{R}_{++}$ as the set of (strict) positive real numbers.

To begin with, we introduce some definitions, which are useful for the following analysis.

**Definition 2.1** ($\alpha$-Lipschitzness). Function $g(x) : X \to \mathbb{R}$ is $\alpha$-Lipschitz ($\alpha \in \mathbb{R}_+$) on $x \in X \subseteq \mathbb{R}^d$, if

$$|g(x) - g(y)| \leq \alpha \|x - y\|$$

for all $x, y \in X$.

**Definition 2.2** ($c$-concavity). A differentiable function $g(x) : X \to \mathbb{R}$ is $c$-(strongly) concave ($c \in \mathbb{R}_+$) on $x \in X \subseteq \mathbb{R}^d$, if $X$ is a convex set and,

$$g(y) \leq g(x) + \langle y - x, \nabla g(x) \rangle - \frac{c}{2} \|y - x\|^2, \quad \forall x, y \in X. \quad (1)$$

Intuitively, we may understand the definition to be that $g(\cdot)$ has a directional curvature less than or equal to $-c$ at any point $x$ inside the convex set $X$ to any direction $y - x$.

**Definition 2.3** ($\alpha$-scaled Pseudo-Gradient Ascend Dynamic). Let $\{u_i(\boldsymbol{x})\}_{i \in [n]}$ be the utility functions of players in an $n$-player game and $\boldsymbol{x}(0)$ be an arbitrary initial strategy profile. We consider a reasonable dynamic of players' strategies, called $\boldsymbol{\alpha}$-scaled pseudo-gradient ascent dynamic, that describes the players' behaviors

over time. The $\boldsymbol{\alpha}$-scaled pseudo-gradient ascent dynamic $\boldsymbol{x}(t) = (x_1(t), \ldots, x_n(t))$ with updating speed $\boldsymbol{\alpha} \in \mathbb{R}_{++}^n$ (possibly $\boldsymbol{\alpha} \neq \mathbf{1}$) is a system of differential equations, defined as

$$\frac{dx_i}{dt}(t) = \alpha_i \frac{\partial u_i}{\partial x_i}(\boldsymbol{x}(t)), \quad \forall i \in [n].$$

Here the vector $(\frac{\partial u_i}{\partial x_i}(\boldsymbol{x}))_{i \in [n]}$ is called the pseudo-gradient of the game $(u_i(\boldsymbol{x}))_{i \in [n]}$ at the point $\boldsymbol{x}$. In this dynamic, each player updates her strategy, taking the direction as the gradient of her utility, scaled by vector $\boldsymbol{\alpha}$.

We finally present a property of $c$-concavity function for use in the subsequent theorems.

**Lemma 2.1.** Assume $g : X \to \mathbb{R}$, where $X \subseteq \mathbb{R}^d$ is a convex and closed set, and $g$ is a differential $c_0$-concave function. Define $x^*$ be the maximum point of $g(x)$ on $X$, then,

$$2c_0 \left(g(x^*) - g(x)\right) \leq \|\nabla g(x)\|^2 \quad \forall x \in X.$$

For completeness, we present self-contained proofs for all results in this paper. Many proofs are deferred to Appendix A due to space limit.

## 2.1 Welfare Solutions

We first consider the social optimal solution. We characterize this concept by social welfare $SW(\boldsymbol{x})$, which is the sum of all players' utilities: $SW(\boldsymbol{x}) = \sum_{i \in [n]} (f_i(k_i) - c_i(x_i))$. The social optimal solution is the effort profile $\boldsymbol{x}^*$ that maximizes $SW(\boldsymbol{x})$.

Since the concavity of social welfare function and the convex, bounded and closed domain $X$, the social optimal solution is guaranteed to exist. However, The social optimal solution may not have an explicit expression, which motivates us to explore the gradient flow as a dynamic process to achieve the social optimal solution:

$$\frac{dx_i}{dt}(t) = \frac{\partial SW}{\partial x_i}(\boldsymbol{x}(t)), \quad i \in [n]. \quad (2)$$

It is well-established that gradient flow converges to a stable point, and in the case of a concave function, any stable point corresponds to a global maximum. Specifically, we have following theorem:

**Theorem 2.2.** The best-response dynamic Equation (2) converges to the social optimal solution with linear rate, i.e.,

$$SW(\boldsymbol{x}^*) - SW(\boldsymbol{x}(t)) \leq \frac{c}{t}, \quad \forall t > 0$$

for some $c > 0$.

Moreover, if at least one of following conditions holds:

(1) all cost functions $c_i(x)$ are $c_0$-convex for some $c_0 > 0$;
(2) all value functions $f_i(k)$ are $c_0$-concave for some $c_0 > 0$;

then, the best-response dynamic converges to the social optimal solution with exponential rate, i.e.,

$$SW(\boldsymbol{x}^*) - SW(\boldsymbol{x}(t)) = O(\exp(-c \cdot t))$$

for some $c > 0$.

Theorem 2.2 establishes the pseudo-gradient ascent dynamic with homogeneous utility function ($SW(\boldsymbol{x})$ in this case), regarded as a continuous-time algorithm, converges to the social optimal point. Though not surprising, the technical insight in this proof are helpful for the later proof on the uniqueness of NE.

---

[1]Since $X_i$ is bounded for all $i$, $\underline{k}_i$ and $\bar{k}_i$ are well-defined. In fact, we have the explicit expression that $\underline{k}_i = \sum_{j \in [n]} \mathbf{1}\{w_{ij} > 0\} w_{ij}\underline{x}_j + \mathbf{1}\{w_{ij} < 0\} w_{ij}\bar{x}_j$ and $\bar{k}_i = \sum_{j \in [n]} \mathbf{1}\{w_{ij} > 0\} w_{ij}\bar{x}_j + \mathbf{1}\{w_{ij} < 0\} w_{ij}\underline{x}_j$

# 3 EQUILIBRIUM SOLUTIONS OF PUBLIC GOOD GAMES

In this section we establish the existence and uniqueness results of equilibrium solution in public good games. We define an effort profile $x$ as a (pure strategy) Nash Equilibrium (NE), if no player can unilaterally increase her utility by changing her effort. Formally, $x$ is an NE if, for any player $i$ and any alternative effort $x_i'$, we have

$$u_i(x_i', x_{-i}) \le u_i(x_i, x_{-i}) \quad \forall x_i' \in [\underline{x}_i, \bar{x}_i]. \tag{3}$$

## 3.1 Existence of Nash Equilibrium

Generally, an (pure) NE may not exist in normal-form games. However, in the networked public good games studied in this paper, we apply the Brouwer's fixed-point theorem [Brouwer 1911] to show that a NE always exists in following theorems.

**Theorem 3.1.** *In the public good game $G = (\{f_i\}_{i \in [n]}, \{c_i\}_{i \in [n]}, \{X_i\}_{i \in [n]}, W)$, an (pure strategy) NE always exists.*

PROOF SKETCH OF THEOREM 3.1. Let us construct the best-response function $\mathrm{BR}(x)$ for all players. If it is continuous, then by Brouwer's fixed point theorem [Brouwer 1911], there is a fixed-point, which is also the NE of the game.

However, this is not always the case, since the best-response may be discontinuous and even not a singleton set. To resolve this issue, we modify the cost function with an $\alpha$-convex function, which ensures that the best-response function of the $\alpha$-modified game is continuous.

As long as the $\alpha$-modified game has a NE $x_\alpha^*$, we let $\alpha \to 0$, then by compactness of $X$, we know that there is an accumulation point $x^*$ that is the limitation of $x_{\alpha_k}^*$ for a sequence $\alpha_k \to 0$. The next step is to prove that $x^*$ is the NE of original game, given $x_{\alpha_k}^*$ is the NE of $\alpha_k$-modified game for all $k$, which needs a few steps in taking limits. □

Similar to normal form games, the Nash Equilibrium (NE) in public goods games may not be unique. This is elaborated upon in the example below.

**Example 1.** *Consider a public good game containing four players, see Figure 1-(a). The marginal gain is $1$ between one player from left side and the other from right side, and $0$ otherwise. We specify homogeneous utility functions and action spaces for all players. The action space is specified as $[0, 1]$, while the only two constraints for utility functions are:*

$$f'(1) \ge c'(1) \text{ and } f'(2) \le c'(0).$$

*It's straightforward to verify that players on one side exerting full effort, i.e., $x_i = 1$, while players on the opposite side free ride, i.e., $x_i = 0$, constitutes a Nash Equilibrium (NE). Thus, there are at least two NEs in this game. This example can be readily extended to a scenario involving $n_1 \times n_2$ players, distributed into $n_1$ groups with $n_2$ players in each group. All pairs of players from different groups are connected, see Figure 1-(b). The second condition then becomes $f'(n_2) \le c'(0)$.*

Example 1 motivates us to investigate the scenarios in which the NE is unique.

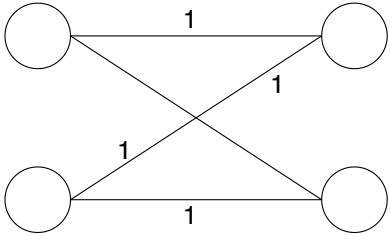

(a)

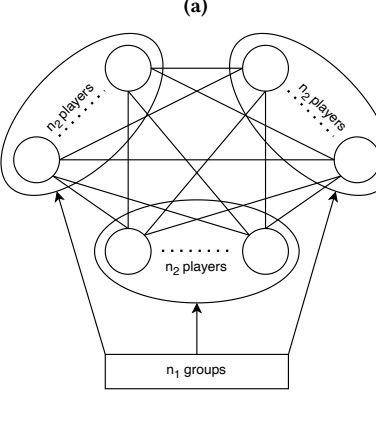

(b)

**Figure 1: Examples of non-unique NE in public good games. (a): There are four players on two sides, with two players in each side. (b) There are $n_1 \times n_2$ players in $n_1$ groups, with $n_2$ players in each group.**

## 3.2 Uniqueness of Nash Equilibrium

In this section, we explore the conditions under which the Nash Equilibrium (NE) of the public good game is unique. We begin by introducing a necessary lemma along with some definitions that will be frequently utilized in the subsequent theorems.

**Definition 3.1** (($\gamma, \sigma$)-closeness). We call a function $g(x) : X \to \mathbb{R}$ is ($\gamma, \sigma$)-close ($\gamma, \sigma \in \mathbb{R}_+$) to a function $f(x) : X \to \mathbb{R}$, where $X \subseteq \mathbb{R}^k$, if $\gamma \nabla_x g(x) - \nabla_x f(x)$ is $\sigma$-Lipschitz on $x$.

**Definition 3.2** (Near-potential Game). Consider a game containing $n$ players. Denote $x_i \in X_i \subset \mathbb{R}$ as the action of player $i$ and $u_i(x)$ as the utility function of player $i$ given joint action $x$.

We say that this game is an ($\gamma, \Sigma$)-near-potential ($\gamma \in \mathbb{R}_{++}^n, \Sigma \in \mathbb{R}_+^{n \times n}$) game w.r.t. a potential function $u(x)$, if for two players $i, j \in [n]$ (it could be $i = j$), we have that $u_i(x)$ is ($\gamma_i, \sigma_{ij}$)-close to $u(x)$ on the domain $X_j$, assuming that $x_{-j}$ is fixed, *i.e.*,

$$\gamma_i \frac{\partial u_i}{\partial x_i}(x_j, x_{-j}) - \frac{\partial u}{\partial x_i}(x_j, x_{-j})$$

is $\sigma_{ij}$-Lipschitz on $x_j$ for all fixed $x_{-j} \in X_{-j}$, where $\Sigma = \{\sigma_{ij}\}_{i,j \in [n]}$.

**Lemma 3.2.** *For a ($\gamma, \Sigma$)-near-potential game w.r.t. potential function $u(x)$, containing $n$ players, if the following holds*

(1) *$u(x)$ is $c$-strongly concave on $x$;*

(2) $c > \sigma_{max}(\Sigma)$,

where $\sigma_{max}(\cdot)$ represents the maximum singular value of a matrix, then the near-potential game has a unique NE $x^*$. Moreover, the $\gamma$-scaled pseudo-gradient ascent dynamic $x(t)$ with arbitrary initial point $x(0)$ converges to the NE with exponential rate, i.e., there is $c_0 > 0$ such that

$$\|x(t) - x^*\| = O(\exp(-c_0 \cdot t)).$$

Lemma 3.2 can be deduced from the results in Rosen [1965] by the utilization of *concave games* and *diagonal strict concavity*, with a technical assumption of the second-order differentiability of $u_i(x)$'s and $u(x)$. The proof is done by verifying the conditions in Rosen [1965] hold given conditions in this lemma. We present the proof sketch in the following and the full derivation is moved to appendix.

Proof Sketch of Lemma 3.2. Rosen [1965] showed that diagonal strict concavity indicates the uniqueness of Nash equilibrium and a sufficient condition for diagonal strict concavity is that $G(x, \gamma) + G^T(x, \gamma)$ is negative definite, where $G(x, \gamma)$ is the Jacobian of $g(x, \gamma)$ w.r.t. $x$, $G^T$ is the transpose of matrix $G$, and $g(x, \gamma)$ is the vector $(\gamma_i \frac{\partial u_i}{\partial x_i}(x))_{i \in [n]}$ representing the pseudo-gradient of game $(u_i(x))_{i \in [n]}$.

By careful computation, we can write that

$$G(x, \gamma) = H(x) + \begin{bmatrix} \frac{\partial^2 (\gamma_1 u_1 - u)}{\partial x_1^2}(x) & \cdots & \frac{\partial^2 (\gamma_1 u_1 - u)}{\partial x_1 \partial x_n}(x) \\ \vdots & \ddots & \vdots \\ \frac{\partial^2 (\gamma u_n - u)}{\partial x_n x_1}(x) & \cdots & \frac{\partial^2 (\gamma u_n - u)}{\partial x_n \partial x_n}(x) \end{bmatrix}$$
$$\triangleq H(x) + I(x, \gamma)$$

where $H(x)$ is the Hessian matrix of $u(x)$ w.r.t. $x$, thus is $c$-negative definite. By near-potential property of game $(u_i(x))_{i \in [n]}$, we can bound the $I(x, \gamma)$ by $\Sigma$, with the largest eigenvalue of $\Sigma + \Sigma^T$ less than $2c$. Therefore, $G(x, \gamma) + G^T(x, \gamma)$ is bound to be negative definite, which completes the proof. □

Next we will present three results of the uniqueness of NE under different conditions.

**Theorem 3.3.** *Given a public goods game* $G = \langle \{f_i(k)\}_{i \in [n]}, \{c_i(x)\}_{i \in [n]}, \{X_i\}_{i \in [n]}, W \rangle$. *If the following conditions hold,*

(1) $\gamma_i (f_i(x + d) - c_i(x))$ is $c$-concave on $x$, for all $i$ and any fixed $d \in [\underline{d}_i, \bar{d}_i] \coloneqq D_i$, where $\underline{d}_i$ and $\bar{d}_i$ are the minimum and maximum externality gains of player $i$, respectively; [2]

(2) $f'_i(k)$ is $L_0$-Lipschitz on $k$ for all $i$;

(3) $c > L_0 \sigma_{max}(\Sigma)$, where $\Sigma = \{\sigma_{ij}\}_{i,j \in [n]}$ and $\sigma_{ij} = \sum_{k \neq i} \gamma_k |w_{ki} w_{kj}|$,

*then, the NE is unique.*

Proof. We shall apply Lemma 3.2 to prove this theorem.

Firstly, we will construct a near-potential game by specifying the potential $u(x)$ and the utilities $u_i(x)$ for all players.

To do this, we let the potential $u(x) = \sum_{i \in [n]} \gamma_i (f_i(k_i) - c_i(x_i))$, and specify the utilities $u_i(x)$ in near-potential game identical with

---

[2]Similar with $\underline{k}_i$ and $\bar{k}_i$, we have the explicit formula as follows: $\underline{d}_i = \sum_{j \neq i} \mathbf{1}\{w_{ij} > 0\} w_{ij} \underline{x}_j + \mathbf{1}\{w_{ij} < 0\} w_{ij} \bar{x}_j$ and $\bar{d}_i = \sum_{j \neq i} \mathbf{1}\{w_{ij} > 0\} w_{ij} \bar{x}_j + \mathbf{1}\{w_{ij} < 0\} w_{ij} \underline{x}_j$

the utilities in the public good game. With some straightforward calculations, we derive that

$$\frac{\partial u_i}{\partial x_i}(x) = f'_i(k_i) - c'_i(x_i)$$

$$\frac{\partial u}{\partial x_i}(x) = \sum_{i' \neq i} \gamma_{i'} f'_{i'}(k_{i'}) w_{i'i} + \gamma_i \left( f'_i(k_i) - c'_i(x_i) \right)$$

Since $f'_{i'}(k_{i'})$ is $L_0$-Lipschitz on $k_{i'}$, and $k_{i'}$ is $|w_{i'j}|$-Lipschitz on $x_j$, we have $\gamma_{i'} f'_{i'}(k_{i'})$ is $L_0 \gamma_{i'} |w_{i'j}|$ Lipschitz on $x_j$ and $\sum_{i' \neq i} \gamma_{i'} f'_{i'}(k_{i'}) w_{i'i}$ is $L_0 \sum_{i' \neq i} \gamma_{i'} |w_{i'j} w_{i'i}|$ Lipschitz on $x_j$.

Construct the matrix $\Sigma = \{\sigma_{ij}\}_{i,j \in [n]}$ with $\sigma_{ij} = L_0 \sum_{k \neq i} \gamma_k |w_{ki} w_{kj}|$, we know that $u_i(x)$ is $\Sigma$-near-potential respect to $u(x)$.

By Lemma 3.2, we obtain the result and thus complete the proof. □

*Remark* 3.1. The conditions in Theorem 3.3 intuitively means that the players are close to play an individual-interest game, *i.e.*, the non-diagonal elements of $W$—those describes the interactions among different players—are small enough. In fact, from the expression of potential $u(x) = \sum_{i \in [n]} \gamma_i u_i(x)$, we know that the NE solution is close to the (weighted) social optimal solution.

**Theorem 3.4.** *Given a public goods game* $G = \langle \{f_i(k)\}_{i \in [n]}, \{c_i(x)\}_{i \in [n]}, \{X_i\}_{i \in [n]}, W \rangle$. *If the following conditions hold,*

(1) $f_i(k)$ is $(\gamma_i, \sigma_i)$-close to $f(k)$ for all $i \in [n]$;

(2) $f(x + d) - \gamma_i c_i(x)$ is $c$-strongly concave on $x$ for all $i \in [n]$ and all $d \in [\underline{d}_i, \bar{d}_i]$, and $f'(k)$ is $c^1$-Lipschitz on $k$, $f''(k)$ is $c^2$-Lipschitz on $k$, $c, c^1, c^2 \in \mathbb{R}_+$;

(3) $c > \sigma_{max}(B)$, where $B = \{\beta_{ij}\}_{i,j \in [n]}$ and $\beta_{ij} = \sigma_i |w_{ij}| + c^1 |w_{ij} - 1| + c^2 \sum_{j \in [n]} |w_{ij} - 1| \max\{-\underline{x}_j, \bar{x}_j\}$,

*then the NE is unique.*

*Remark* 3.2. It's important to note that the conditions specified in Theorem 3.4 intuitively suggest that each element of $W$ closely approximates 1, and the values derived from the gains $f_i(k)_{i \in [n]}$ are nearly identical (when scaled). Consequently, the game approaches the characteristics of an identical-interest game, where the players' actions nearly maximize the potential function $u(x) = f(\|x\|_1) - \sum_{i \in [n]} \gamma_i c_i(x_i)$. However, the social welfare is close to $n f(\|x\|_1) - \sum_{i \in [n]} \gamma_i c_i(x_i)$, the $\frac{1}{n}$ coefficients on values means that in this case, the free-ride phenomenon can occur.

**Theorem 3.5.** *Given a public goods game* $G = \langle \{f_i(k)\}_{i \in [n]}, \{c_i(x)\}_{i \in [n]}, \{X_i\}_{i \in [n]}, W \rangle$. *If the following conditions hold,*

(1) $W^0$ is positive definite and $\sigma_{min}(W^0) = \sigma_0 > 0$. We also restrict $w_{ii}^0 = 1$, $\forall i \in [n]$ where $W^0 = \{w_{ij}^0\}_{i,j \in [n]}$;

(2) $c'_i(x)$ is $L_i$-Lipschitz on $x$ for all $i$;

(3) $f_i(k)$ is $C_i$-concave on $k$ for all $i$;

(4) $\sigma_0 > \sigma_{max}(\Sigma)$, where $\Sigma = \{\sigma_{ij}\}_{i,j \in [n]}$ and $\sigma_{ii} = 0$ and $\sigma_{ij} = \frac{2L_i |w_{ij}|}{C_i} + |w_{ij}^0 - w_{ij}|$,

*where $\sigma_{min}(W)$ represents the minimal eigenvalue of a symmetric matrix $W$, then the NE is unique.*

Proof Sketch of Theorem 3.5. Bayer et al. [2023] proved that, when $W$ is symmetric and the cost functions $c_i(x)$s are linear, then the best-response dynamic converges. The insight is that when $c_i(x)$s are linear, each player $i$ has its own marginal cost $c_i$, and

the ideal $k_i$ such that $f'_i(k_i) = c_i$. Therefore, every player $i$ plays the best-response to her ideal gain $k_i$, and $\phi(\boldsymbol{x}) = \boldsymbol{k}^T \boldsymbol{x} - \frac{1}{2} \boldsymbol{x}^T W \boldsymbol{x}$ becomes a potential function. Moreover, the NE must be unique if $W$ is positive semi-definite.

Our proof follows this insight. We construct the potential $\phi(\boldsymbol{x}) = \boldsymbol{k}^{*T} \boldsymbol{x} - \frac{1}{2} \boldsymbol{x}^T W^0 \boldsymbol{x}$. Similarly define $y_i(\boldsymbol{x}_{-i})$ as the optimal gain level of player $i$, when the strategy profile of other players is $\boldsymbol{x}_{-i}$. Then the utilities in the near-potential game are constructed as,

$$\phi_i(\boldsymbol{x}) = y_i(\boldsymbol{x}_{-i})x_i - \frac{x_i^2}{2} - \sum_{j \neq i} w_{ij} x_i x_j.$$

We then need to verify that: (1) the NE of near-potential game corresponds to the NE of original public good game; and (2) the constructed game $\{\phi_i(\boldsymbol{x})\}_{i \in [n]}$ is indeed a near-potential game. We completes the proof by Lemma 3.2. □

*Remark 3.3.* Theorem 3.5 intuitively suggests that, if $W$ is close to a positive definite matrix $W_0$, as well as that the profit functions $f_i(k)$s are more concave than cost functions $c_i(x)$s, then the NE could be unique.

In addition to these three theorems that ensure the uniqueness of the Nash Equilibrium (NE), we also explore a concept known as game equivalence, which can expand the applicability of these theorems.

**Definition 3.3** (Game Equivalence). Given two public goods games $G^1, G^2$ with $n$ players, where

$$G^j = (\{f_i^j\}_{i \in [n]}, \{c_i^j\}_{i \in [n]}, \{X_i^j = [\underline{x}_i^j, \bar{x}_i^j]\}_{i \in [n]}, W^j), j \in \{1, 2\}.$$

We say that $G^1$ is equivalent to $G^2$, if there is a diagonal matrix $D = \operatorname{diag}(d_1, ..., d_n)$, $d_i \in \mathbb{R}_{++}$ and an offset vector $\boldsymbol{b} \in \mathbb{R}^n$, satisfying that,

$$W^2 = DW^1 D^{-1}$$
$$\underline{x}_i^2 = d_i \underline{x}_i^1 + b_i$$
$$\bar{x}_i^2 = d_i \bar{x}_i^1 + b_i$$
$$c_i^1(x) = c_i^2(d_i x + b_i) \quad \forall x \in X_i^1$$
$$f_i^1(k) = f_i^2(d_i k + m_i) \quad \forall k \in K_i^1$$

where $m_1, ..., m_n$ are constants such that $m_i = d_i \sum_{j \in [n]} \frac{w_{ij}^1 b_j}{d_j}$.

Intuitively, Definition 3.3 says that, if $G^1$ is equivalent to $G^2$, then $G^1$ and $G^2$ are intrinsically same in terms of linear transformation. Through this insight, we have following theorem.

**Theorem 3.6.** *If two games, $G_1$ and $G_2$, are equivalent, then there exists a one-to-one mapping between NEs of $G_1$ and the NEs of $G_2$.*

From Theorem 3.6, we can easily know that the uniqueness property of NE keep the same under equivalent class. Therefore, we have corollary below, which can further broaden the class of public goods game with unique NE.

**Corollary 3.7.** *For a public goods game $G^1$, if $G^1$ is equivalent to game $G^2$, and $G^2$ satisfies the conditions in Theorem 3.3, Theorem 3.4 or Theorem 3.5, then $G^1$ has a unique NE.*

# 4 CASE STUDY

## 4.1 Comparative Statics: Money Redistribution for Welfare Analysis

In this section, we study comparative statics, *i.e.*, how the players' utilities will change if the model parameters are modified by an infinitesimal amount. We characterize the infinitesimal modification by money redistribution, *i.e.*, replace $\{f_i(k_i)\}_{i \in [n]}$ by $\{f_i(k_i + \delta_i t)\}_{i \in [n]}$, where $\boldsymbol{\delta} = (\delta_1, \ldots, \delta_n) \in \mathbb{R}^n$ is called the direction of money redistribution and $t \in \mathbb{R}$ is called the change magnitude. Overall, there is an $\boldsymbol{\delta} t$ shift in the gain level of players. The goal of infinitesimal change drive us to study the case $t \to 0$.

In this way, the utility of player $i$ becomes,

$$u_i(\boldsymbol{x}; t) = f_i(k_i + \delta_i t) - c_i(x_i),$$

Denote $\boldsymbol{x}^*(t)$ as the NE when the change magnitude is $t$. We do not assume the uniqueness of NE anymore, and $\boldsymbol{x}^*(t)$ might be not unique. However, we assume the first-order differentiability of $\boldsymbol{x}^*(t)$ with respect to $t$, as well as that $\boldsymbol{x}^*(t)$ is an inner point of $X$. These assumptions are quite natural. For the first assumption, if the game changes with an infinitesimal magnitude and players always achieve the rational outcome, *i.e.*, NE, then it is imaginable and intuitive that the outcome of players should also change minimally. The second assumption is only technical. We denote $u_i(t) = u_i(\boldsymbol{x}^*(t); t)$ for a little abuse of notation when the context is clear. We mainly concern about $u_i'(0)$, which means that what the marginal change of $\delta$ would have effect on the players utilities.

We build this result as follows.

**Theorem 4.1.** *Assume $u_i(t)$ and $\boldsymbol{x}^*(t)$ are defined above, and denote $\boldsymbol{x}^* = \boldsymbol{x}^*(0)$, $\boldsymbol{k}^* = W\boldsymbol{x}^*$, then,*

$$\boldsymbol{u}'(0) = \operatorname{diag}(\boldsymbol{f}'(\boldsymbol{k}^*)) \cdot \operatorname{diag}(\boldsymbol{c}''(\boldsymbol{x}^*) - \boldsymbol{f}''(\boldsymbol{k}^*))$$
$$\left(\operatorname{diag}(\boldsymbol{c}''(\boldsymbol{x}^*)) - W\operatorname{diag}(\boldsymbol{f}''(\boldsymbol{k}^*))\right)^{-1} \boldsymbol{\delta}$$

*where $\boldsymbol{u}(0)$ represents the utility profile $(u_1(0), u_2(0), \ldots, u_n(0))$.*

We show some examples to illustrate the implication of this result.

**Example 2.** *Here are some simple cases of Theorem 4.1.*

(1) *If the value function is linear on gain, i.e., $f_i''(k) \equiv 0$, then, it becomes that*

$$\boldsymbol{u}'(0) = \operatorname{diag}(\boldsymbol{f}'(\boldsymbol{k}^*))\boldsymbol{\delta}$$

*This result is pretty intuitive, since a linear value function indicates that the NE is unique and is constant, because the marginal values for players' efforts are constants and marginal costs only depend on players' own value. Therefore, the change of money redistribution has a direct change on the utilities.*

(2) *If the cost function is linear on effort, i.e., $c_i''(x) \equiv 0$, then, it becomes that*

$$\boldsymbol{u}'(0) = \operatorname{diag}(\boldsymbol{f}'(\boldsymbol{k}^*))W^{-1}\boldsymbol{\delta}$$

*In this case, NE might be not constant and not unique (see Example 1). Therefore, the redistribution of money will affect the interactions of players, thus have an indirect effect on the utilities. Specifically, the indirect effect imposes the inverse of*

$W$—the matrix that portrays the interactions of players—to the money redistribution $\delta$.

(3) If we want the money redistribution to be Pareto dominant, i.e., $u_i'(0) \geq 0$ for all players, since the first two diagonal matrices are positive diagonal matrices, the only requirements of $\delta$ is:

$$\left[\text{diag}(c''(x^*)) - W\text{diag}(f''(k^*))\right]^{-1} \delta \geq 0$$

Besides, a linear cost would reduce the requirements to,

$$W^{-1}\delta \geq 0$$

## 4.2 Some Applications of Results

In this section, we propose a specific example to illustrate how the results regarding the uniqueness of Nash equilibrium (NE) can be applied in practice. This example is inspired by Fershtman and Nitzan [1991], where the cost functions are modeled as quadratic functions.

Specifically, we assume the homogeneity of players in the public good game $G$, i.e., the values of gains, costs of efforts, and action spaces are identical among players, with differences only in the network structure $W$. Therefore, we use $f(k)$ and $c(x)$ instead of $f_i(k_i)$ and $c_i(x_i)$ to represents values and costs, when the context allows.

Assume $f(k)$ and $c(x)$ has following expression:

$$f(k) = \begin{cases} ak - bk^2 & \text{if } 0 \leq k \leq \frac{a}{2b} \\ \frac{a^2}{4b} & \text{if } k > \frac{a}{2b} \end{cases}$$

$$c(x) = \frac{c_0}{2}x^2 \qquad \text{for } c_0 > 0$$

and $X = [0, \bar{x}]$ for a sufficiently large $\bar{x}$ such that choosing $\bar{x}$ is a dominated strategy for all players, due to extremely high costs and bounded values for gains. The values and costs are quadratic functions in their domains, with a clipping on value function at the maximum point. We also restrict $w_{ij}$ to be either 0 or 1.

From the expressions of $f(k)$ and $c(x)$, we know that $c(x)$ is $c_0$-strongly convex, $c'(x)$ is $c_0$-Lipschitz, $f(k)$ is $2b$-strongly concave in the domain $[0, \frac{a}{2b}]$ and $f'(k)$ is $2b$-Lipschitz on the full domain.

### 4.2.1 The Application of Theorem 3.3.
In this part, we assume that the non-diagonal elements of $W$ is i.i.d. generated with probability $p = \frac{p_0}{n}$ equals to 1 and 0 otherwise, where $p_0 > 0$ is a constant. We have following theorem,

**Theorem 4.2.** if $\frac{c_0}{2b} > 2p_0 + p_0^2 + \sqrt{n(8p_0 + 10p_0^2 + 4p_0^3)}$, then with probability at least $\frac{1}{2}$, the public good game $G$ has a unique NE.

*Proof Sketch and Remark.* This proof is done by substituting Theorem 3.3 and using Chebyshev's inequalities. Notice that $\sigma_{max}(\Sigma)$ can be bounded by the $\infty$-norm $\|\Sigma\|_\infty$, which is the maximum row sum of $\Sigma$. We extract sum of each row $i$ by $\gamma_i$, using Chebyshev's inequalities to bound the tail of $\gamma_i$ and union bound to control $\|\Sigma\|_\infty = \max_i \gamma_i$.

Notice that the result inevitably has a dependency on the square root of $n$ by Chebshev's inequality. Due to dependence between $\sigma_{ij}$ and $\sigma_{ij'}$, we can not directly use concentration inequalities, such as Chernoff's inequality [Chernoff 1952], which can help decrease the dependency to $\log n$. However, we believe that the poly $\log(n)$

dependency can be established, by the intrinsic independence on $\{w_{ij}\}_{i,j \in [n]}$, which allows for further studies.

### 4.2.2 The Application of Theorem 3.5 and Theorem 3.6.
In this part, we assume that $W$ has a specific up-triangular structure, i.e., $w_{ij} = 0$ if $i > j$. Next we will show that under this assumption, the NE of public good game is unique.

**Theorem 4.3.** If $W$ is an up-triangular matrix, i.e., $w_{ij} = 0$ for $i > j$, then the public good game $G$ has unique NE.

*Proof Sketch and Remark.* In such scenarios, the conditions specified in Theorems 3.3 to 3.5 may no longer be satisfied. However, we can employ the technique described in Theorem 3.6 to transform the original game $G$ into another game $G'$ that meets the conditions outlined in Theorem 3.5.

Notice that this game must have unique NE. It is because the following insight: since $w_{ij}$ for $i > j$ means that the efforts of players with lower identifiers $j$ have no externalities on players with higher identifiers $i$. Therefore, player $n$ is playing an individual-interest game, thus have an optimal strategy $x_n^*$. Given $x_n^*$ fixed, player $n-1$ can also determine an optimal strategy $x_{n-1}^*$. Overall, each player can determine an optimal strategy in turn, which forms an equilibrium. However, our proof can give a stronger results that, if $w_{ij} = O(\varepsilon^{i+1-j})^3$ for $i > j$, we can also guarantee the uniqueness of NE. [4]

## 5 CONCLUSION

In this paper, we have presented a novel approach to understanding networked public goods games featuring heterogeneous players and convex cost functions. Through rigorous analysis and theoretical explorations, we have expanded the conventional understanding of strategic interactions in public goods provision within networked environments. Our model, which integrates heterogeneous benefits and convex costs, provides a more realistic portrayal of individual contributions and the resultant dynamics compared to traditional models with linear and homogeneous cost structures.

The theoretical insights and methodological contributions of our study on networked public goods games with heterogeneous players and convex costs fill a significant gap in the economic theory, and also provide new perspectives for policymakers. Specifically, by understanding the conditions under which Nash Equilibrium can be achieved and sustained, policymakers can better design interventions and incentives in the context of the Internet economy and social networks, that encourage optimal contribution levels to public goods. In future research, it would be valuable to extend this model to consider dynamic environments, where players can adjust their strategies over time. Additionally, incorporating stochastic elements to model uncertainty in player interactions could provide further insights into the robustness of the model in more complex and realistic scenarios.

---

[3]here $\varepsilon$ is a constant used in the proof
[4]It is because after the transformation in the proof, the lower-triangular elements hold to be $O(\varepsilon)$.

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

## A OMITTED PROOFS

### A.1 Proof of Lemma 2.1

**Lemma 2.1.** *Assume $g : X \rightarrow \mathbb{R}$, where $X \subseteq \mathbb{R}^d$ is a convex and closed set, and $g$ is a differential $c_0$-concave function. Define $x^*$ be the maximum point of $g(x)$ on $X$, then,*

$$2c_0\left(g(x^*) - g(x)\right) \leq \|\nabla g(x)\|^2 \quad \forall x \in X.$$

PROOF. The case $c_0 = 0$ is trivial. Consider $c_0 > 0$, take maximum on both sides of Equation (1), we have

$$g(x^*) = \max_{y \in X} \quad g(y)$$

$$\leq \max_{y \in \mathbb{R}^d} \quad g(x) + \langle y - x, \nabla g(x) \rangle - \frac{c_0}{2}\|y - x\|^2$$

The maximum of RHS is achieved at $y^* = x + \frac{1}{c_0}\nabla g(x)$, and therefore,

$$g(x^*) \leq g(x) + \langle y^* - x, \nabla g(x) \rangle - \frac{c_0}{2}\|y^* - x\|^2$$

$$= g(x) + \frac{1}{2c_0}\|\nabla g(x)\|^2$$

which completes the proof. □

### A.2 Proof of Theorem 2.2

**Theorem 2.2.** *The best-response dynamic Equation (2) converges to the social optimal solution with linear rate, i.e.,*

$$SW(x^*) - SW(x(t)) \leq \frac{c}{t}, \quad \forall t > 0$$

*for some $c > 0$.*

*Moreover, if at least one of following conditions holds:*

*(1) all cost functions $c_i(x)$ are $c_0$-convex for some $c_0 > 0$;*
*(2) all value functions $f_i(k)$ are $c_0$-concave for some $c_0 > 0$;*

*then, the best-response dynamic converges to the social optimal solution with exponential rate, i.e.,*

$$SW(x^*) - SW(x(t)) = O(\exp(-c \cdot t))$$

*for some $c > 0$.*

PROOF. *Case 1:* We first consider the case that one of the conditions hold. If $c_i(x_i)$ is $c_0$-convex, then we know that $SW(x)$ is $c_0$-convex on $x_i$. Similarly, if $f_i(k_i)$ is $c_0$-concave, since $k_i$ depends linearly on $x_i$, we also know that $SW(x)$ is $c_0$-convex on $x_i$. Also if $SW(x)$ is $c_0$-concave on $x_i$ for all $i$, then $SW(x)$ is $c_0$-concave on $x$.

As a property of $c_0$-concave function $f(x)$ and maximum point $x^*$, we have

$$f(x^*) - f(x) \leq \frac{1}{2c_0}\|\nabla f(x)\|^2. \tag{4}$$

Define the energy function $E(t) = SW(x^*) - SW(x(t))$, then

$$\frac{dE(t)}{dt} = -\sum_{i \in [n]} \frac{\partial SW}{\partial x_i}(x(t))\frac{dx_i}{dt}(t) = -\sum_{i \in [n]}\left(\frac{\partial SW}{\partial x_i}(x(t))\right)^2 = -\|\nabla SW(x(t))\|^2.$$

Since $SW(x)$ is $c_0$-concave, by Equation (4) we have $\|\nabla SW(x(t))\|^2 \geq 2c_0(SW(x^*) - SW(x)) = 2c_0 E(t)$. Therefore, we have

$$\frac{dE(t)}{dt} \leq -2c_0 E(t).$$

By standard differential equation analysis, we have $E(t) \leq \exp(-2c_0 t)E(0)$. Taking $c = -2c_0$ completes the proof.

*Case 2:* Next, we consider the general case. Define $J(t) = t(SW(x^*) - SW(x(t))) + \frac{1}{2}\|x^* - x(t)\|^2$. We have

$$\frac{dJ(t)}{dt} = SW(x^*) - SW(x(t)) - t\langle \frac{\partial SW}{\partial x}(x(t)), \frac{dx}{dt} \rangle - \langle x^* - x(t), \frac{dx}{dt} \rangle$$

$$= SW(x^*) - SW(x(t)) - \langle x^* - x(t), \nabla SW(x(t)) \rangle - t\|\nabla SW(x(t))\|^2$$

By concavity of $SW(x)$ we have $SW(x^*) - SW(x(t)) \leq \langle x^* - x(t), \nabla SW(x(t)) \rangle$, then we have $\frac{dJ(t)}{dt} \leq 0$. This indicates that

$$J(t) \leq J(0) = \frac{1}{2}\|x^* - x(0)\|^2$$

where $J(t) \geq t(\mathrm{SW}(\boldsymbol{x}^*) - \mathrm{SW}(\boldsymbol{x}(t)))$, and therefore

$$\mathrm{SW}(\boldsymbol{x}^*) - \mathrm{SW}(\boldsymbol{x}(t)) \leq \frac{1}{2t} \|\boldsymbol{x}^* - \boldsymbol{x}(0)\|^2$$

Taking $c = \frac{\|\boldsymbol{x}^* - \boldsymbol{x}(0)\|^2}{2}$ completes the proof.

□

## A.3 Proof of Theorem 3.1

**Theorem 3.1.** *In the public good game* $G = (\{f_i\}_{i \in [n]}, \{c_i\}_{i \in [n]}, \{X_i\}_{i \in [n]}, W)$, *an (pure strategy) NE always exists.*

PROOF. We first consider the case where all $c_i(x_i)$s are $c$-strongly concave for some $c > 0$, in which case best responses of players are always unique and continuous. We will generalize the result to the general case in the second step.

*Case 1: Strongly Convex Cost.* Consider the best response function of players: $\mathrm{BR} : \times_{i \in [n]} X_i \to \times_{i \in [n]} X_i$, where $\mathrm{BR}_i(\boldsymbol{x})$ represents the best response of player $i$ given the effort profile $\boldsymbol{x}_{-i}$, omitting the dummy variable $x_i$.

Consider the utility function of player $i$:

$$u_i(\boldsymbol{x}) = f_i\left(\sum_{j \in [n]} w_{ij} x_j\right) - c_i(x_i)$$

$$\mathrm{BR}_i(\boldsymbol{x}) = \arg\max_{x_i'} u_i(x_i', \boldsymbol{x}_{-i})$$

Now we will show the continuity of $\mathrm{BR}_i(\boldsymbol{x})$. We assume the negation holds. It indicates that there is two sequence $\{\boldsymbol{x}_k^j\}_{k \in \mathbb{N}_+}, j \in \{1, 2\}$ such that $\lim \boldsymbol{x}_k^1 = \lim \boldsymbol{x}_k^2$ but $\lim \mathrm{BR}_i(\boldsymbol{x}_k^1) \neq \lim \mathrm{BR}_i(\boldsymbol{x}_k^2)$ or one of the limitations do not exist. If the latter hold, since the compactness of $X$ we can choose a sub-sequence of $\{\boldsymbol{x}_k^j\}$ such that the limitation of $\mathrm{BR}_i(\boldsymbol{x}_k^j)$ exists for $j = 1, 2$. Therefore we only consider the former case.

Let $d = \|\lim \mathrm{BR}_i(\boldsymbol{x}_k^1) - \lim \mathrm{BR}_i(\boldsymbol{x}_k^2)\|$. By optimality of $\mathrm{BR}_i(\boldsymbol{x}_k^j)$, we have

$$u_i(\mathrm{BR}_i(\boldsymbol{x}_k^j), \boldsymbol{x}_{k,-i}^j) \geq u_i(\mathrm{BR}_i(\boldsymbol{x}_k^{-j}, \boldsymbol{x}_{k,-i}^j) + \frac{c}{2}\|\mathrm{BR}_i(\boldsymbol{x}_k^j) - \mathrm{BR}_i(\boldsymbol{x}_k^{-j})\|^2, \quad j = 1, 2$$

Sum up with $j = 1, 2$, we have

$$\sum_{j=1}^2 \left[ u_i(\mathrm{BR}_i(\boldsymbol{x}_k^j), \boldsymbol{x}_{k,-i}^j) - u_i(\mathrm{BR}_i(\boldsymbol{x}_k^j), \boldsymbol{x}_{k,-i}^{-j}) \right] \geq c\|\mathrm{BR}_i(\boldsymbol{x}_k^1) - \mathrm{BR}_i(\boldsymbol{x}_k^2)\|^2$$

Then taking limits of $k \to \infty$, we know that LHS becomes 0 and RHS becomes $cd^2 > 0$, which leads a contradiction.

Since $\times_{i \in [n]} X_i$ is a bounded convex set, by Brouwer's fixed-point theorem, we know that there exists $\boldsymbol{x} \in X$ such that $\mathrm{BR}(\boldsymbol{x}) = \boldsymbol{x}$, which indicates that $\boldsymbol{x}$ is an NE.

*Case 2: General Convex Cost.* To deal with the case that $c_i(x_i)$ might be not strongly concave, we use the technique of utility reshaping. Specifically, we define another public good game $G^\beta = (\{f_i\}_{i \in [n]}, \{c_i^\beta\}_{i \in [n]} \{X_i\}_{i \in [n]}, W)$ where $\beta > 0$ and $c^\beta(x_i) = c(x_i) + \beta x_i^2$. It's obvious that in public good game $G^\beta$, the cost functions are $\beta$-strongly concave, and the NE must exist.

We denote an NE of $G^\beta$ as $\boldsymbol{x}^\beta$. The next step is to construct a strategy profile $\boldsymbol{x}$, from $\boldsymbol{x}^\beta$ for $\beta > 0$, such that $\boldsymbol{x}$ is an NE of $G$. Notice that a simple limit may not work, since there is no guarantee that $\boldsymbol{x}^\beta$ is continuous with $\beta$, even that $\boldsymbol{x}^\beta$ might be unmeasurable with $\beta$.

To resolve this issue, we notice that $\boldsymbol{x}^\beta \in X$ where $X$ is a compact set. By the Bolzano-Weierstrass theorem, we know that there exists a convergent subsequence $\boldsymbol{x}^{\beta_k} \to \boldsymbol{x}$ for some $\boldsymbol{x} \in X$ and $\beta_k \xrightarrow{k \to \infty} 0, k \in \mathbb{Z}_+$.

Finally we verify the NE property of $\boldsymbol{x}$. Notice that $X$ is compact again, we know that $c^{\beta_k}(x_i)$ converges to $c(x_i)$ consistently, therefore, $u_i^\beta(\boldsymbol{x}) = f_i(k_i) - c_i^{\beta_k}(x_i)$ also converges to $u_i(\boldsymbol{x})$ consistently. Take limit on both sides of following equality,

$$u_i^{\beta_k}(\boldsymbol{x}^{\beta_k}) = \max_{x_i' \in X_i} u_i^{\beta_k}(x_i', \boldsymbol{x}_{-i}^{\beta_k}),$$

we achieve that,

$$u_i(\boldsymbol{x}) = \max_{x_i' \in X_i} u_i(x_i', \boldsymbol{x}_{-i}),$$

which indicates that $\boldsymbol{x}$ is an NE of $G$.

□

## A.4   The derivation of Lemma 3.2 with the results of concave games in Rosen [1965]

**Lemma 3.2.** *For a $(\gamma, \Sigma)$-near-potential game w.r.t. potential function $u(x)$, containing $n$ players, if the following holds*

    (1) $u(x)$ *is $c$-strongly concave on $x$;*

    (2) $c > \sigma_{max}(\Sigma)$,

*where $\sigma_{max}(\cdot)$ represents the maximum singular value of a matrix, then the near-potential game has a unique NE $x^*$. Moreover, the $\gamma$-scaled pseudo-gradient ascent dynamic $x(t)$ with arbitrary initial point $x(0)$ converges to the NE with exponential rate, i.e., there is $c_0 > 0$ such that*

$$\|x(t) - x^*\| = O(\exp(-c_0 \cdot t)).$$

PROOF. To prove Lemma 3.2, we only need to check the *diagonal strictly concavity* of the public good game.

Denote $g(x, \gamma) = (\gamma_1 \cdot \frac{\partial u_1}{\partial x_1}(x), \ldots, \gamma_n \cdot \frac{\partial u_n}{\partial x_n}(x))$ and $G(x, \gamma)$ be the Jacobian of $g(x, \gamma)$ with respect to $x$. Rosen [1965] shows that if $(G(x, \gamma) + G^T(x, \gamma))$ is negative definite for all $x$ where $G^T$ represents the transpose of $G$, then the original game must be diagonal strictly concave.

Now we compute the $G(x, \gamma)$ directly.

$$G(x, \gamma) = \begin{bmatrix} \gamma_1 \cdot \frac{\partial^2 u_1}{\partial x_1^2}(x) & \cdots & \gamma_1 \cdot \frac{\partial^2 u_1}{\partial x_1 \partial x_n}(x) \\ \vdots & \ddots & \vdots \\ \gamma_n \cdot \frac{\partial^2 u_n}{\partial x_n x_1}(x) & \cdots & \gamma_n \cdot \frac{\partial^2 u_n}{\partial x_n \partial x_n}(x) \end{bmatrix}$$

Let $H(x)$ be the Hessian matrix of $u(x)$. We have that

$$G(x, \gamma) = H(x) + \begin{bmatrix} \frac{\partial^2 (\gamma_1 u_1 - u)}{\partial x_1^2}(x) & \cdots & \frac{\partial^2 (\gamma_1 u_1 - u)}{\partial x_1 \partial x_n}(x) \\ \vdots & \ddots & \vdots \\ \frac{\partial^2 (\gamma u_n - u)}{\partial x_n x_1}(x) & \cdots & \frac{\partial^2 (\gamma u_n - u)}{\partial x_n \partial x_n}(x) \end{bmatrix}$$

$$\triangleq H(x) + I(x, \gamma)$$

By $c$-concavity of $u(x)$, we know that $H(x) + H^T(x)$ is negative definite with largest eigenvalue smaller than $-2c$.

By $(\gamma, \Sigma)$ near-potential property of the public good game, we know that the term $|\frac{\partial (\gamma_i u_i - u)}{\partial x_i \partial x_j}(x)| \leq \sigma_{ij}$. Therefore, the maximum eigenvalue of $I(x, \gamma) + I^T(x, \gamma)$ can be upper bounded by the maximum eigenvalue of $\Sigma + \Sigma^T$, which is also upper bounded by $2\sigma_{max}(\Sigma)$.

Above all, for all $x \neq 0$,

$$x^T G(x, \gamma) x = x^T H(x) x + x^T I(x, \gamma) x$$

$$\leq -c\|x\|^2 + \|x^T\|\|I(x, \gamma)x\|$$

$$\leq -c\|x\|^2 + \|x\|\sigma_{max}(I(x, \gamma))\|x\|$$

$$\leq -c\|x\|^2 + \|x\|\sigma_{max}(\Sigma)\|x\|$$

$$= (\sigma_{max}(\Sigma) - c)\|x\|^2$$

$$< 0$$

Therefore, the public good game is diagonal strictly concave, and we completes the proof.

□

## A.5   A Self-contained Proof of Lemma 3.2

PROOF. *Step 1: The uniqueness of NE.*

We prove the uniqueness of NE mainly by constructing a contraction mapping and using Banach fixed-point theorem. The contraction mapping is constructed by descritized version of best-response dynamic:

$$x_i' = g_i(x) = x_i + \varepsilon \gamma_i \frac{\partial u_i}{\partial x_i}(x_i, x_{-i}), \quad \forall i \in [n]$$

for some $\varepsilon > 0$ small enough. Now we prove that $g : X \to X$ is a contraction mapping.

Consider two effort profiles $x, y \in X$, we have that

$$\|g(x) - g(y)\|^2 = \sum_{i \in [n]} (g_i(x) - g_i(y))^2$$

$$= \sum_{i \in [n]} \left( x_i + \varepsilon \gamma_i \frac{\partial u_i}{\partial x_i}(x_i, x_{-i}) - y_i - \varepsilon \gamma_i \frac{\partial u_i}{\partial x_i}(y_i, y_{-i}) \right)^2$$

$$= \|x - y\|^2 + \varepsilon \left\langle x - y, \gamma_i \left( \frac{\partial u_i}{\partial x_i}(x_i, x_{-i}) - \frac{\partial u_i}{\partial x_i}(y_i, y_{-i}) \right)_{i \in [n]} \right\rangle$$

$$+ \varepsilon^2 \sum_{i \in [n]} \gamma_i^2 \left( \frac{\partial u_i}{\partial x_i}(x_i, x_{-i}) - \frac{\partial u_i}{\partial x_i}(y_i, y_{-i}) \right)^2$$

We focus on $\varepsilon$ term:

$$\left\langle x - y, \gamma_i \left( \frac{\partial u_i}{\partial x_i}(x_i, x_{-i}) - \frac{\partial u_i}{\partial x_i}(y_i, y_{-i}) \right)_{i \in [n]} \right\rangle$$

$$= \left\langle x - y, \frac{\partial u}{\partial x}(x) - \frac{\partial u}{\partial x}(y) \right\rangle + \left\langle x - y, \left( \frac{\partial (\gamma_i u_i - u)}{\partial x_i}(x_i, x_{-i}) - \frac{\partial (\gamma_i u_i - u)}{\partial x_i}(y_i, y_{-i}) \right)_{i \in [n]} \right\rangle$$

Since the $c$-concavity of $u(x)$, we have

$$\left\langle x - y, \frac{\partial u}{\partial x}(x) - \frac{\partial u}{\partial x}(y) \right\rangle \leq -c\|x - y\|^2$$

By $\sigma_{ij}$-Lipschitzness of $\frac{\partial(\gamma_i u_i - u)}{\partial x_i}(x)$ on $x_j$, we have

$$\left\langle x - y, \left( \frac{\partial (\gamma_i u_i - u)}{\partial x_i}(x_i, x_{-i}) - \frac{\partial (\gamma_i u_i - u)}{\partial x_i}(y_i, y_{-i}) \right)_{i \in [n]} \right\rangle$$

$$\leq \sum_{i \in [n]} \left( \frac{\partial (\gamma_i u_i - u)}{\partial x_i}(x_i, x_{-i}) - \frac{\partial (\gamma_i u_i - u)}{\partial x_i}(y_i, y_{-i}) \right) |x_i - y_i|$$

$$\leq \sum_{i \in [n]} \sum_{j \in [n]} \sigma_{ij}|x_j - y_j||x_i - y_i|$$

$$= \|x - y\|^2 \sum_{i \in [n]} \sum_{j \in [n]} \sigma_{ij} z_i z_j$$

$$= \|x - y\|^2 z^T \Sigma z$$

$$\leq \sigma_{max}(\Sigma)\|x - y\|^2$$

where in the third equality, $z_i := \frac{|x_i - y_i|}{\|x - y\|}$, we have $\|z_i\| = 1$, notice that $\sigma_{max}(\Sigma) = \max_{\|z\| = \|z'\| = 1} z^T \Sigma z'$.

Therefore, the $\varepsilon$ term:

$$\left\langle x - y, \gamma_i \left( \frac{\partial u_i}{\partial x_i}(x_i, x_{-i}) - \frac{\partial u_i}{\partial x_i}(y_i, y_{-i}) \right)_{i \in [n]} \right\rangle$$

$$\leq (\sigma_{max} - c)\|x - y\|^2$$

Therefore, we can choose $\varepsilon$ so small such that $\|g(x) - g(y)\|^2 \leq (1 + \frac{\varepsilon}{2}(\sigma_{max} - c))\|x - y\|^2 < \|x - y\|^2$ for all $x, y$, which indicates that $g$ is a contraction mapping. By Banach fixed-point theorem, we know that there exists a unique fixed point $x^*$ of $g$, which means that there is a unique $x^*$ such that $\frac{\partial u_i}{\partial x_i}(x^*) = 0$ for all $i$, indicating that $x^*$ is the unique NE.

*Step 2: The exponential convergence rate.*

To do this, we aim at constructing an energy function $E(x)$ such that it holds with the following properties:

- $E(x) \geq 0$ for all $x$, the equality holds if $x = x^*$.
- $\frac{dE(x(t))}{dt} \leq -p_0 E(x(t))$ for some $c_0 > 0$.

As long as these properties hold, we immediately get that $E(x(t)) \leq E(x(0)) \exp(-p_0 t)$, take $c_0 = p_0$ completes the proof.

We first define the energy function $E(x) = u(x^*) - u(x) + \langle x - x^*, \nabla u(x^*) \rangle$. Since $u(x)$ is a concave function, we know that,

$$u(y) - u(x) \leq \langle y - x, \nabla u(x) \rangle$$

Take $x = x^*$ and $y = x$, we derive that $E(x) \geq 0$ for all $x$. When $x = x^*$, we have $E(x^*) = 0$. We also know that $E(x)$ is $c$-strongly convex function.

By little computation and define $v_i(\boldsymbol{x}) = \gamma_i u_i(\boldsymbol{x}) - u(\boldsymbol{x})$,

$$\frac{\partial E}{\partial \boldsymbol{x}}(\boldsymbol{x}) = \left( -\frac{\partial u}{\partial \boldsymbol{x}}(\boldsymbol{x}) + \frac{\partial u}{\partial \boldsymbol{x}}(\boldsymbol{x}^*) \right)$$

$$\frac{\mathrm{d} x_i}{\mathrm{d} t}(t) = \gamma_i \frac{\partial u_i}{\partial x_i}(\boldsymbol{x}(t)) = \frac{\partial v_i}{\partial x_i}(\boldsymbol{x}(t)) + \frac{\partial u}{\partial x_i}(\boldsymbol{x}(t)) = \frac{\partial v_i}{\partial x_i}(\boldsymbol{x}(t)) + \frac{\partial u}{\partial x_i}(\boldsymbol{x}^*) - \frac{\partial E}{\partial x_i}(\boldsymbol{x}(t))$$

Next, we compute the derivative of $E(\boldsymbol{x}(t))$:

$$\frac{\mathrm{d} E}{\mathrm{d} t}(\boldsymbol{x}(t)) = \left\langle \frac{\partial E}{\partial \boldsymbol{x}}(\boldsymbol{x}(t)), \frac{\mathrm{d} \boldsymbol{x}}{\mathrm{d} t}(t) \right\rangle$$

$$= -\left\| \frac{\partial E}{\partial \boldsymbol{x}}(\boldsymbol{x}(t)) \right\|^2 \cdots \text{first term}$$

$$+ \left\langle \frac{\partial E}{\partial \boldsymbol{x}}(\boldsymbol{x}(t)), \frac{\partial u}{\partial \boldsymbol{x}}(\boldsymbol{x}^*) \right\rangle + \sum_{i \in [n]} \frac{\partial v_i}{\partial x_i}(\boldsymbol{x}(t)) \frac{\partial E}{\partial x_i}(\boldsymbol{x}(t)) \cdots \text{second term}$$

We also know $\frac{\partial u_i}{\partial x_i}(\boldsymbol{x}^*) = 0$ by definition of NE. Combining them on the second term, we achieve,

$$\text{second term} = \sum_{i \in [n]} \frac{\partial E}{\partial x_i}(\boldsymbol{x}(t)) \cdot \frac{\partial u}{\partial x_i}(\boldsymbol{x}^*) + \sum_{i \in [n]} \frac{\partial v_i}{\partial x_i}(\boldsymbol{x}(t)) \cdot \frac{\partial E}{\partial x_i}(\boldsymbol{x}(t))$$

$$= \sum_{i \in [n]} \frac{\partial E}{\partial x_i}(\boldsymbol{x}(t)) \cdot \frac{\partial (u - \gamma_i u_i)}{\partial x_i}(\boldsymbol{x}^*) + \sum_{i \in [n]} \frac{\partial v_i}{\partial x_i}(\boldsymbol{x}(t)) \cdot \frac{\partial E}{\partial x_i}(\boldsymbol{x}(t))$$

$$= -\sum_{i \in [n]} \frac{\partial E}{\partial x_i}(\boldsymbol{x}(t)) \cdot \frac{\partial v_i}{\partial x_i}(\boldsymbol{x}^*) + \sum_{i \in [n]} \frac{\partial v_i}{\partial x_i}(\boldsymbol{x}(t)) \cdot \frac{\partial E}{\partial x_i}(\boldsymbol{x}(t))$$

$$= \sum_{i \in [n]} \frac{\partial E}{\partial x_i}(\boldsymbol{x}(t)) \cdot \left( \frac{\partial v_i}{\partial x_i}(\boldsymbol{x}(t)) - \frac{\partial v_i}{\partial x_i}(\boldsymbol{x}^*) \right)$$

Denote $\Delta v_i = \frac{\partial v_i}{\partial x_i}(\boldsymbol{x}(t)) - \frac{\partial v_i}{\partial x_i}(\boldsymbol{x}^*)$ and $\Delta \boldsymbol{v} = (\Delta v_1, ..., \Delta v_n)$, we have,

$$\text{second term} = \left\langle \frac{\partial E}{\partial \boldsymbol{x}}(\boldsymbol{x}(t)), \Delta \boldsymbol{v} \right\rangle$$

$$\leq \left\| \frac{\partial E}{\partial \boldsymbol{x}}(\boldsymbol{x}(t)) \right\| \|\Delta \boldsymbol{v}\|$$

We also know that $\frac{\partial v_i}{\partial x_i}$ is $\sigma_{ij}$-Lipschitz on $x_j$, now we consider $\|\Delta v_i\|$,

$$\|\Delta v_i\| = \max_{\|z\|=1} \sum_{i \in [n]} z_i \Delta v_i$$

$$\leq \max_{\|z\|=1} \sum_{i \in [n]} \sum_{j \in [n]} z_i \sigma_{ij} |x_j(t) - x_j^*|$$

$$\leq \|\boldsymbol{x}(t) - \boldsymbol{x}^*\| \max_{\|z\|=1, \|y\|=1} \sum_{i \in [n]} \sum_{j \in [n]} \sigma_{ij} z_i y_j$$

$$= \|\boldsymbol{x}(t) - \boldsymbol{x}^*\| \sigma_{max}(\Sigma)$$

$$\leq \frac{\sigma_{max}(\Sigma)}{c} \left\| \frac{\partial E}{\partial \boldsymbol{x}}(\boldsymbol{x}(t)) \right\|$$

Combining these, we have,

$$\frac{\mathrm{d} E}{\mathrm{d} t}(\boldsymbol{x}(t)) \leq -\left( 1 - \frac{\sigma_{max}(\Sigma)}{c} \right) \left\| \frac{\partial E}{\partial \boldsymbol{x}}(\boldsymbol{x}(t)) \right\|^2$$

$$\leq -2(c - \sigma_{max}(\Sigma)) E(\boldsymbol{x}(t))$$

Take $p_0 = 2(c - \sigma_{max}(\Sigma))$, we complete the proof.

$\square$

## A.6 Proof of Theorem 3.4

**Theorem 3.4.** *Given a public goods game* $G = \langle \{f_i(k)\}_{i \in [n]}, \{c_i(x)\}_{i \in [n]}, \{X_i\}_{i \in [n]}, W \rangle$. *If the following conditions hold,*

(1) $f_i(k)$ *is* $(\gamma_i, \sigma_i)$-*close to* $f(k)$ *for all* $i \in [n]$;

(2) $f(x + d) - \gamma_i c_i(x)$ *is* $c$-*strongly concave on* $x$ *for all* $i \in [n]$ *and all* $d \in [\underline{d}_i, \bar{d}_i]$, *and* $f'(k)$ *is* $c^1$-*Lipschitz on* $k$, $f''(k)$ *is* $c^2$-*Lipschitz on* $k$, $c, c^1, c^2 \in \mathbb{R}_+$;

(3) $c > \sigma_{max}(B)$, *where* $B = \{\beta_{ij}\}_{i,j \in [n]}$ *and* $\beta_{ij} = \sigma_i |w_{ij}| + c^1 |w_{ij} - 1| + c^2 \sum_{j \in [n]} |w_{ij} - 1| \max\{-\underline{x}_j, \bar{x}_j\}$,

*then the NE is unique.*

PROOF. consider the potential function $u(\boldsymbol{x}) = f(\sum_{i \in [n]} x_i) - \sum_{i \in [n]} \gamma_i c_i(x_i)$. We have that $u(\boldsymbol{x})$ is $c$-strongly concave on $x_i$ for all $i$, thus $c$-strongly concave on $\boldsymbol{x}$.

We also derive that,

$$\frac{\partial u}{\partial x_i}(\boldsymbol{x}) = f'\Big(\sum_{i \in [n]} x_i\Big) - \gamma_i c_i'(x_i)$$

$$\gamma_i \frac{\partial u_i}{\partial x_i}(\boldsymbol{x}) = \gamma_i f_i'\Big(x_i + \sum_{j \neq i} w_{ij} x_j\Big) - \gamma_i c_i'(x_i)$$

$$\frac{\partial(\gamma_i u_i - u)}{\partial x_i}(\boldsymbol{x}) = \gamma_i f_i'\Big(x_i + \sum_{j \neq i} w_{ij} x_j\Big) - f'\Big(\sum_{i \in [n]} x_i\Big)$$

$$= \gamma_i f_i'\Big(x_i + \sum_{j \neq i} w_{ij} x_j\Big) - f'\Big(x_i + \sum_{j \neq i} w_{ij} x_j\Big) \cdots \text{first term}$$

$$+ f'\Big(\sum_{j \in [n]} w_{ij} x_j\Big) - f'\Big(\sum_{i \in [n]} x_i\Big) \cdots \text{second term}$$

For the first term, since $f_i(k)$ is $(\gamma_i, \sigma_i)$-close to $f(k)$, we know that $\gamma_i f_i'(x_i + \sum_{j \neq i} w_{ij} x_j) - \gamma f'(x_i + \sum_{j \neq i} w_{ij} x_j)$ is $\sigma_i |w_{ij}|$-Lipschitz on $x_j$ for all $j$.

For the second term, by Lipschitzness of $f'(k)$ and $f''(k)$ and some computations, consider two points $x_j$ and $x_j + \delta$,

$$\Big| \Big( f'\Big(\sum_{j \in [n]} w_{ij} x_j\Big) - f'\Big(\sum_{i \in [n]} x_i\Big) \Big) - \Big( f'\Big(\sum_{k \neq j} w_{ik} x_k + w_{ij}(x_j + \delta)\Big) - f'\Big(\sum_{k \neq j} x_k + (x_j + \delta)\Big) \Big) \Big|$$

$$= \Big| \Big( f'\Big(\sum_{j \in [n]} w_{ij} x_j\Big) - f'\Big(\sum_{j \in [n]} w_{ij} x_j + \delta\Big) \Big) - \Big( f'\Big(\sum_{i \in [n]} x_i\Big) - f'\Big(\sum_{i \in [n]} x_i + \delta\Big) \Big)$$

$$+ f'\Big(\sum_{j \in [n]} w_{ij} x_j + \delta\Big) - f'\Big(\sum_{j \in [n]} w_{ij} x_j + w_{ij}\delta\Big) \Big|$$

$$\leq \Big| \Big( f'\Big(\sum_{j \in [n]} w_{ij} x_j\Big) - f'\Big(\sum_{j \in [n]} w_{ij} x_j + \delta\Big) \Big) - \Big( f'\Big(\sum_{i \in [n]} x_i\Big) - f'\Big(\sum_{i \in [n]} x_i + \delta\Big) \Big) \Big| \cdots \text{first term}$$

$$+ \Big| f'\Big(\sum_{j \in [n]} w_{ij} x_j + \delta\Big) - f'\Big(\sum_{k \in [n]} w_{ik} x_k + w_{ij}\delta\Big) \Big| \cdots \text{second term}$$

The first term is upper bounded by $\delta \sum_{j \in [n]} |w_{ij} - 1| \max\{-\underline{x}_j, \bar{x}_j\} c^2$, while the second term is upper bounded by $|w_{ij} - 1| \delta c^1$. Overall, this term is $c^1 |w_{ij} - 1| + c^2 \sum_{j \in [n]} |w_{ij} - 1| \max\{-\underline{x}_j, \bar{x}_j\}$-Lipschitz on $x_j$.

Above all, $u_i$ is $(\gamma_i, \beta_{ij})$-close to $u$ on $x_j$, where

$$\beta_{ij} = \sigma_i |w_{ij}| + c^1 |w_{ij} - 1| + c^2 \sum_{j \in [n]} |w_{ij} - 1| \max\{-\underline{x}_j, \bar{x}_j\}$$

Therefore, we completes the proof by Lemma 3.2.

□

## A.7 Proof of Theorem 3.5

**Theorem 3.5.** *Given a public goods game* $G = \langle \{f_i(k)\}_{i \in [n]}, \{c_i(x)\}_{i \in [n]}, \{X_i\}_{i \in [n]}, W \rangle$. *If the following conditions hold,*

(1) $W^0$ *is positive definite and* $\sigma_{min}(W^0) = \sigma_0 > 0$. *We also restrict* $w_{ii}^0 = 1$, $\forall i \in [n]$ *where* $W^0 = \{w_{ij}^0\}_{i,j \in [n]}$;

(2) $c_i'(x)$ is $L_i$-Lipschitz on $x$ for all $i$;

(3) $f_i(k)$ is $C_i$-concave on $k$ for all $i$;

(4) $\sigma_0 > \sigma_{max}(\Sigma)$, where $\Sigma = \{\sigma_{ij}\}_{i,j\in[n]}$ and $\sigma_{ii} = 0$ and $\sigma_{ij} = \frac{2L_i|w_{ij}|}{C_i} + |w_{ij}^0 - w_{ij}|$,

where $\sigma_{min}(W)$ represents the minimal eigenvalue of a symmetric matrix $W$, then the NE is unique.

PROOF. Bayer et al. [2023] shows that, when $W$ is symmetric and the cost functions $c_i(x)$s are linear, then the best-response dynamic converges. The insight is that when $c_i(x)$s are linear, each player $i$ has its own marginal cost $c_i$, and the ideal $k_i$ such that $f_i'(k_i) = c_i$. Therefore, every player $i$ best-response to her ideal gain $k_i$, and $\phi(x) = k^T x - \frac{1}{2}x^T W x$ becomes a potential function. Moreover, the NE must be unique if $W$ is positive semi-definite.

Our proof follows this insight, and try to utilize the conclusion of Lemma 3.2. By Theorem 3.1 we know that there is an NE $x^*$, let $k^* = Wx^*$ be the gain profile in the equilibrium level. We construct the potential following,

$$u(x) = k^{*T}x - \frac{1}{2}x^T W_0 x$$

It's easy to observe that $u(x)$ is $\sigma_0$-strongly concave.

Now consider $y_i(x_{-i})$ as the best response function of player $i$, i.e., $y_i(x_{-i})$ is the gain level $k_i$ such that, it's optimal for player $i$ to choose the effort level $x_i$ such that her gain level becomes $k_i$. If we define $u_i^0(k_i, x_{-i})$ is the utility of player $i$ when other players play $x_{-i}$ and the gain level of player $i$ is $k_i$, we can write that,

$$u_i^0(k_i, x_{-i}) = f_i(k_i) - c_i(k_i - \sum_{j \neq i} w_{ij}x_j)$$

$$= (-c_i^0(k_i, x_{-i})) + f_i(k_i)$$

where $c_i^0(k_i, x_{-i}) = c_i(k_i - \sum_{j\neq i} w_{ij}x_j)$ is the cost function of player $i$ (in another form).

We have $c_i'(k_i)$ is $L_i$-Lipschitz on $k_i$ by assumption, therefore, we could eaasily find that $\frac{\partial^2 c_i^0}{\partial k_i \partial x_j}(k_i, x_{-i})$ is upper bounded by $|w_{ij}|L_i$. Togerther with $f_i(k_i)$ is $C_i$-concave on $k_i$, by ??, we have that

$$y_i(x_{-i}) = \arg\max_{k_i} \quad u_i^0(k_i, x_{-i}) = (-c_i^0(k_i, x_{-i})) + f_i(k_i)$$

is $\frac{2L_i|w_{ij}|}{C_i}$-Lipschitz on $x_j$.

Now we define the utility function for player $i$ in the near-potential game,

$$u_i(x) = y_i(x_{-i})x_i - \frac{x_i^2}{2} - \sum_{j\neq i} w_{ij}x_i x_j$$

If $x$ is an NE for the game $\{u_i(x)\}_{i\in[n]}$ constructed above, we have that

$$0 = \frac{\partial u_i}{\partial x_i}(x) = y_i(x_{-i}) - x_i - \sum_{j\neq i} w_{ij}x_j \tag{5}$$

$$\Rightarrow y_i(x_{-i}) = x_i + \sum_{j\neq i} w_{ij}x_j \tag{6}$$

i.e., the choice of $x_i$ will make her gain level to $y_i(x_{-i})$, which is also the optimal gain level of player $i$ given $x_{-i}$ by definition of $y(x_{-i})$, therefore, $x$ is also an NE of the original public good game $G$.

The last step is to show that the NE for the near-potential game $\{u_i(x)\}_{i\in[n]}$ is unique. We derive that,

$$\frac{\partial u}{\partial x_i}(x) = k_i^* - \sum_{j\in[n]} w_{ij}^0 x_j$$

$$\frac{\partial u_i}{\partial x_i}(x) = y_i(x_{-i}) - \sum_{j\in[n]} w_{ij}x_j$$

It's obvious that $\frac{\partial(u_i-u)}{\partial x_i}(x)$ is $\sigma_{ii} = |w_{ii}^0 - 1| = 0$-Lipschitz on $x_i$ (constant on $x_i$) and $\sigma_{ij} = \frac{2L_i|w_{ij}|}{C_i} + |w_{ij}^0 - w_{ij}|$-Lipschitz on $x_j$. Thus the constructed utilities, $\{u_i(x)\}_{i\in[n]}$, are $(1, \Sigma)$-near potential to $u(x)$. By Lemma 3.2, we have that the NE of the game $\{u_i(x)\}_{i\in[n]}$ is unique, which completes the proof. □

## A.8 Proof of Theorem 3.6

**Theorem 3.6.** *If two games, $G_1$ and $G_2$, are equivalent, then there exists a one-to-one mapping between NEs of $G_1$ and the NEs of $G_2$.*

PROOF. We prove this theorem by reduction, *i.e.*, there is a injective function $g : X^1 \to X^2$ such that if $\boldsymbol{x}^1$ is an NE in $G^1$, then $\boldsymbol{x}^2$ is an NE in $G^2$.

We construct $\boldsymbol{x}^2$ by follows,

$$x_i^2 = d_i x_i^1 + b_i$$

The construction is obviously injective. Therefore, we have

$$k_i^2 = \sum_{j \in [n]} w_{ij}^2 x_j^2 = \sum_{j \in [n]} \frac{d_i}{d_j} w_{ij}^1 (d_j x_j^1 + b_j)$$

$$= d_i \sum_{j \in [n]} w_{ij}^1 x_j^1 + d_1 \sum_{j \in [n]} \frac{w_{ij}^1 b_j}{d_j}$$

$$= d_i k_i^1 + m_i$$

Now fix $\boldsymbol{x}_{-i}^2$, consider the case that player $i$ choose action $x_i^2$:

$$u_i^2(x_i^2, \boldsymbol{x}_{-i}^2) = f_i^2(k_i^2) - c_i^2(x_i^2)$$

$$= f_i^2(d_i k_i^1 + m_i) - c_i^2(d_i x_i^1 + b_i)$$

$$= f_i^1(k_i^1) - c_i^1(x_i^1)$$

which is the maximum utility player $i$ can achieve, since $\boldsymbol{x}^1$ is an NE of $G^1$. Therefore, $\boldsymbol{x}^2$ is an NE of $G^2$.

We also need to prove that the inverse direction also holds, to clarify this statements, we show that the equivalence relation is symmetric, *i.e.*, if $G^1$ is equivalent to $G^2$, then $G^2$ is also equivalent to $G^1$.

To show this, we let $d_i' = 1/d_i$ and $b_i' = -b_i/d_i$, we have $d_i' \in \mathbb{R}_{++}$ and $b_i' \in \mathbb{R}$. Denote $D' = D^{-1} = \text{diag}(d_1', ..., d_n')$, then we have

$$W^1 = D'W^2 D'^{-1}$$

$$\underline{x}_i^1 = d_i' \underline{x}_i^2 + b_i'$$

$$\bar{x}_i^1 = d_i' \bar{x}_i^2 + b_i'$$

$$c_i^2(x) = c_i^1(d_i' x + b_i') \quad \forall x \in X_i^2$$

$$f_i^2(k) = f_i^1(d_i' k + m_i') \quad \forall k \in K_i^2$$

for some constants $\{m_i'\}_{i \in [n]}$. This indicates that an NE of $G^2$ also corresponds to an NE of $G^1$, which completes the proof. □

## A.9 Proof of Theorem 4.1

**Theorem 4.1.** *Assume $u_i(t)$ and $\boldsymbol{x}^*(t)$ are defined above, and denote $\boldsymbol{x}^* = \boldsymbol{x}^*(0)$, $\boldsymbol{k}^* = W\boldsymbol{x}^*$, then,*

$$\boldsymbol{u}'(0) = \text{diag}(\boldsymbol{f}'(\boldsymbol{k}^*)) \cdot \text{diag}(\boldsymbol{c}''(\boldsymbol{x}^*) - \boldsymbol{f}''(\boldsymbol{k}^*))$$

$$(\text{diag}(\boldsymbol{c}''(\boldsymbol{x}^*)) - W\text{diag}(\boldsymbol{f}''(\boldsymbol{k}^*)))^{-1} \boldsymbol{\delta}$$

*where $\boldsymbol{u}(0)$ represents the utility profile $(u_1(0), u_2(0), \ldots, u_n(0))$.*

PROOF. $\boldsymbol{x}^*(t)$ should satisfies,

$$x_i^*(t) = \arg\max_{x_i} f_i(x_i + \sum_{j \neq i} w_{ij} x_j^*(t) + \delta_i t) - c_i(x_i), \tag{7}$$

By Equation (7), we have

$$c_i'(x_i^*(t)) - f_i'(k_i^*(t) + \delta_i t) = 0,$$

which carves out an implicit function

$$F_i(\boldsymbol{x}^*(t), t) = 0, \quad \forall i \in [n]$$

with $F_i(\boldsymbol{x}, t) = c_i'(x_i) - f_i'(k_i + \delta_i t)$. Take $E_i(t) = F_i(\boldsymbol{x}^*(t), t)$, by implicit function theorem, we have

$$\frac{dE_i}{dt}(t) = \frac{\partial F_i}{\partial \boldsymbol{x}}(\boldsymbol{x}^*(t), t) \frac{d\boldsymbol{x}^*}{dt}(t) + \frac{\partial F_i}{\partial t}(\boldsymbol{x}^*(t), t) = 0 \tag{8}$$

Together Equation (8) with all $i$, we have

$$\frac{\mathrm{d}\boldsymbol{x}^*}{\mathrm{d}t}(t) = -\left(\frac{\partial F}{\partial \boldsymbol{x}}\right)^{-1}\frac{\partial F}{\partial t}(\boldsymbol{x}^*(t), t)$$

where $F(\boldsymbol{x}^*(t), t) = (F_1(\boldsymbol{x}^*(t), t), F_2(\boldsymbol{x}^*(t), t), \ldots, F_n(\boldsymbol{x}^*(t), t))$, by computation we have,

$$\frac{\partial F}{\partial t}(\boldsymbol{x}^*(t), t) = -\mathrm{diag}(f''(\boldsymbol{k}^*(t) + t\boldsymbol{\delta}))\boldsymbol{\delta}$$

$$\frac{\partial F}{\partial \boldsymbol{x}}(\boldsymbol{x}^*(t), t) = \mathrm{diag}(c''(\boldsymbol{x}^*(t))) - \mathrm{diag}(f''(\boldsymbol{k}^*(t) + t\boldsymbol{\delta}))W$$

By $\boldsymbol{u}(t) = \boldsymbol{u}(x^*(t); t)$, we derive that,

$$\boldsymbol{u}'(0) = \frac{\partial \boldsymbol{u}}{\partial t}(\boldsymbol{x}^*; 0) + \frac{\partial \boldsymbol{u}}{\partial \boldsymbol{x}}(\boldsymbol{x}^*; 0)\frac{\mathrm{d}\boldsymbol{x}^*}{\mathrm{d}t}(0)$$

where

$$\frac{\partial \boldsymbol{u}}{\partial t}(\boldsymbol{x}^*; 0) = \mathrm{diag}(f'(\boldsymbol{k}^*))\boldsymbol{\delta}$$

$$\frac{\partial \boldsymbol{u}}{\partial \boldsymbol{x}}(\boldsymbol{x}^*; 0) = \mathrm{diag}(f'(\boldsymbol{k}^*))W - \mathrm{diag}(c'(\boldsymbol{x}^*))$$

$$\frac{\mathrm{d}\boldsymbol{x}^*}{\mathrm{d}t}(0) = \left(\mathrm{diag}(c''(\boldsymbol{x}^*)) - \mathrm{diag}(f''(\boldsymbol{k}^*))W\right)^{-1}\mathrm{diag}(f''(\boldsymbol{k}^*))\boldsymbol{\delta}$$

By equilibrium condition, we have

$$\mathrm{diag}(c'(\boldsymbol{x}^*)) = \mathrm{diag}(f'(\boldsymbol{k}^*))$$

and thus

$$\frac{\partial \boldsymbol{u}}{\partial \boldsymbol{x}}(\boldsymbol{x}^*; 0) = \mathrm{diag}(f'(\boldsymbol{k}^*))(W - I)$$

Above all,

$$\boldsymbol{u}'(0) = \mathrm{diag}(f'(\boldsymbol{k}^*))\left(\boldsymbol{\delta} + (W - I)\left(\mathrm{diag}(c''(\boldsymbol{x}^*)) - \mathrm{diag}(f''(\boldsymbol{k}^*))W\right)^{-1}\mathrm{diag}(f''(\boldsymbol{k}^*))\boldsymbol{\delta}\right)$$

$$= \mathrm{diag}(f'(\boldsymbol{k}^*))\left(I + (W - I)\left(\mathrm{diag}(c''(\boldsymbol{x}^*)) - \mathrm{diag}(f''(\boldsymbol{k}^*))W\right)^{-1}\mathrm{diag}(f''(\boldsymbol{k}^*))\right)\boldsymbol{\delta}$$

$$= \mathrm{diag}(f'(\boldsymbol{k}^*))\left(I + (W - I)\left(\mathrm{diag}(c''(\boldsymbol{x}^*)/f''(\boldsymbol{k}^*)) - W\right)^{-1}\right)\boldsymbol{\delta}$$

$$= \mathrm{diag}(f'(\boldsymbol{k}^*))\left(\left(\mathrm{diag}(c''(\boldsymbol{x}^*)/f''(\boldsymbol{k}^*)) - W\right) \cdot \left(\mathrm{diag}(c''(\boldsymbol{x}^*)/f''(\boldsymbol{k}^*)) - W\right)^{-1}\right.$$

$$\left. + (W - I)\left(\mathrm{diag}(c''(\boldsymbol{x}^*)/f''(\boldsymbol{k}^*)) - W\right)^{-1}\right)\boldsymbol{\delta}$$

$$= \mathrm{diag}(f'(\boldsymbol{k}^*))\left(\left(\mathrm{diag}(c''(\boldsymbol{x}^*)/f''(\boldsymbol{k}^*)) - I\right)\left(\mathrm{diag}(c''(\boldsymbol{x}^*)/f''(\boldsymbol{k}^*)) - W\right)^{-1}\right)\boldsymbol{\delta}$$

$$= \mathrm{diag}(f'(\boldsymbol{k}^*)) \cdot \mathrm{diag}(c''(\boldsymbol{x}^*) - f''(\boldsymbol{k}^*)) \cdot \left[\mathrm{diag}(c''(\boldsymbol{x}^*)) - W\mathrm{diag}(f''(\boldsymbol{k}^*))\right]^{-1}\boldsymbol{\delta}$$

which completes the proof.

□

## A.10 Proof of Theorem 4.2

**Theorem 4.2.** *if $\frac{c_0}{2b} > 2p_0 + p_0^2 + \sqrt{n(8p_0 + 10p_0^2 + 4p_0^3)}$, then with probability at least $\frac{1}{2}$, the public good game $G$ has a unique NE.*

Proof. We firstly substitute the model into the conditions of Theorem 3.3.

- For the first condition, we just specify $\gamma_i \equiv 1$, then $f_i(x + d) - c_i(x)$ is $c$-concave.
- For the second condition, $f_i'(k)$ is $2b$-Lipschitz on $k$ for all $i$.
- For the third condition, we need that $\sigma_{max}(\Sigma) < \frac{c}{2b}$, where $\sigma_{ij} = \sum_{k \neq i} w_{ki}w_{kj}$.

It's well-known that $\sigma_{max}(\Sigma) \leq \min\{\|\Sigma\|_\infty, \|\Sigma\|_1\}$, where $\|\Sigma\|_\infty$ ($\|\Sigma\|_1$) represents the $\infty$-norm (one-norm) of $\Sigma$, i.e., maximum row sum (column sum) of $\Sigma$, respectively. Specifically,

$$\|\Sigma\|_\infty = \max_i \sum_{j \in [n]} \sum_{k \neq i} w_{ki} w_{kj}$$

$$= \max_i \sum_{k \neq i} w_{ki} + \sum_{j \neq i} \sum_{k \neq i} w_{ki} w_{kj}$$

$$= \max_i \sum_{k \neq i} w_{ki} + \sum_{j \neq i} w_{ji} + \sum_{j \neq i} \sum_{k \neq i,j} w_{ki} w_{kj}$$

$$= \max_i \quad 2 \sum_{j \neq i} w_{ji} + \sum_{j \neq i} \sum_{k \neq i,j} w_{ki} w_{kj}$$

Denote $\delta_i = 2 \sum_{j \neq i} w_{ij} + \sum_{j \neq i} \sum_{k \neq i,j} w_{ki} w_{kj}$, then,

$$\mathbb{E}_W[\delta_i] = 2(n-1)p + (n-1)(n-2)p^2 \leq 2p_0 + p_0^2$$

When the context is clear we denote $\mathbb{E}_W[\delta] = \mathbb{E}_W[\delta_i]$, since this term is constant.

We also compute the second-order moment of $\delta_i$, i.e.,

$$\delta_i^2 = \left( 2 \sum_{j_1 \neq i} w_{j_1,i} + \sum_{j_1 \neq i, k_1 \neq j_1,i} w_{j_1,i} w_{j_1,k_1} \right)^2$$

$$= 4 \sum_{j_1 \neq i, j_2 \neq i} w_{j_1,i} w_{j_2,i} + 4 \sum_{j_2 \neq i} \sum_{j_1 \neq i, k_1 \neq j_1,i} w_{j_1,i} w_{j_1,k_1} w_{j_2,i} + \sum_{j_1 \neq i, k_1 \neq j_1,i} \sum_{j_2 \neq i, k_2 \neq j_2,i} w_{j_1,i} w_{j_2,i} w_{j_1,k_1} w_{j_2,k_2}$$

By simple counting, we have,

$$\mathbb{E}_W[\delta_i^2] = 4(n-1)p + 9(n-1)(n-2)p^2 + 6(n-1)(n-2)^2 p^3 + (n-1)(n-2)(n^2 - 5n + 5)p^4$$

and the variance of $\delta_i$,

$$\mathrm{Var}_W[\delta_i] = \mathbb{E}_W[\delta_i^2] - \mathbb{E}_W^2[\delta_i]$$

$$= 4(n-1)p + (n-1)(5n-14)p^2 + (n-1)(n-2)(2n-8)p^3 + (n-1)(n-2)(-2n+3)p^4$$

$$\leq 4p_0 + 5p_0^2 + 2p_0^3$$

Combining them by using Chebyshev inequalities,

$$\Pr[\delta_i - \mathbb{E}[\delta] \geq k] \leq \frac{\mathrm{Var}[\delta_i]}{k^2}$$

and

$$\Pr[\|\Sigma\|_\infty - \mathbb{E}[\delta] \geq k] \leq \frac{n \mathrm{Var}[\delta_i]}{k^2}$$

since $\|\Sigma\|_\infty = \max_{i \in [n]} \delta_i$.

To make RHS $= \frac{1}{2}$, we take $k = \sqrt{n(8p_0 + 10p_0^2 + 4p_0^3)}$, therefore, with probability at least $\frac{1}{2}$, we have that $\sigma_{max}(\Sigma) \leq \|\Sigma\|_\infty \leq 2p_0 + p_0^2 + \sqrt{n(8p_0 + 10p_0^2 + 4p_0^3)} < \frac{c}{2b}$, and the game has unique NE by Theorem 3.3.

$\square$

## A.11 Proof of Theorem 4.3

**Theorem 4.3.** *If $W$ is an up-triangular matrix, i.e., $w_{ij} = 0$ for $i > j$, then the public good game $G$ has unique NE.*

PROOF. We denote the original game as $G^1 = \{\{f_i^1\}, \{c_i^1\}, \{X_i^1\}, W^1\}$, and the transformed game as $G^2 = \{\{f_i^2\}, \{c_i^2\}, \{X_i^2\}, W^2\}$. $f_i^1$ is $2b$-concave and $(c_i^1)'$ is $c$-Lipschitz.

To do a game transformation, we need to construct scaling vector $\boldsymbol{d} \in \mathbb{R}_{++}$ and offset vector $\boldsymbol{b} \in \mathbb{R}_{++}$. We set $\boldsymbol{b} = \boldsymbol{0}$ without loss of generality, we have that,

$$w_{ij}^2 = \frac{d_i w_{ij}^1}{d_j}$$

$w_{ij}^2$ also equals to 0 if $i > j$ and 1 if $i = j$, for $i < j$, we observe that if $d_i << d_j$ as long as $j > i$, then $w_{ij}^2$ can be arbitrarily small. In fact, we let $d_i = \varepsilon^{-i}$, where $\varepsilon > 0$ is a pre-specific constant. Therefore, we have that

$$\begin{cases} w_{ij}^2 = 0 & \text{if } i > j \\ w_{ij}^2 = 1 & \text{if } i = j \\ w_{ij}^2 \leq \varepsilon & \text{if } i < j \end{cases}$$

By this transformation, we have that $f_i^2$ is $\frac{2b}{d_i^2}$-concave and $(c_i^2)'$ is $\frac{c}{d_i^2}$-Lipschitz, in Theorem 3.6.

Now we prove that game $G^2$ satisfies the conditions in Theorem 3.5. We choose $W^0 = I$ so that $\sigma_0 = \sigma_{min}(W^0) = 1$. $L_i = \frac{c}{d_i^2}$ and $C_i = \frac{2b}{d_i^2}$. We compose $\Sigma - \Sigma^1 + \Sigma^2$, where $\Sigma^k = \{\sigma_{ij}^k\}$ and $\sigma_{ij}^1 = \frac{2L_i|w_{ij}^2|}{C_i}$, $\sigma_{ij}^2 = |w_{ij}^0 - w_{ij}^2|$. We have $\sigma_{max}(\Sigma) \leq \sigma_{max}(\Sigma^1) + \sigma_{max}(\Sigma^2)$.

Since $|w_{ij}^0 - w_{ij}^2| = 0$ if $i = j$ and $\leq \varepsilon$ is $i \neq j$, we know that $\sigma_{max}(\Sigma^2)$ is bounded by $n\varepsilon$. Besides, $\sigma_{ij}^1 = \frac{2L_i|w_{ij}^2|}{C_i} \leq \frac{c\varepsilon}{b}$, therefore $\sigma_{max}(\Sigma^1)$ is bounded by $\frac{nc\varepsilon}{b}$. Therefore, if we choose $\varepsilon < \frac{b}{n(b+c)}$, we can get $\sigma_{max}(\Sigma) < 1 = \sigma_0$, which indicates that $G^2$ has unique NE, which is same for $G^1$.

□

