# OpenReview forum: "Networked Digital Public Goods Games with Heterogeneous Players and Convex Costs"
_ACM.org/TheWebConf/2025/Conference — WWW 2025 Oral_

### Official Review · Reviewer_R6UW · 2024-11-25

**Novelty:** 5
**Technical Quality:** 7

**Review:**

Digital public goods like open-source software and databases are often under-provided for due to free-riding. Previous work has investigated such “public goods games” under linear cost models with homogeneous playerss. This submission studies networked public good games when players are heterogeneous and their costs are convex.

In the model, each player exerts some effort. Each player’s utility is a concave “benefit” function minus a convex “cost” function (of their effort). The authors show that, in this game, a pure strategy Nash equilibrium always exists. The proof relies on Brouwer’s fixed point theorem. They also provide conditions on the game under which the Nash equilibrium is unique, and show that pseudo-gradient descent dynamics will converge to this point. The authors also introduce the idea of “game equivalence”, which enables the transformation of one public goods game into another while preserving the one-to-one relationships between the Nash equilibria in both games.

Finally, the authors conclude with a case study where they study comparative statics (how player utilities change if the model parameters are modified by a small amount).

While the authors are not the first to study networked public good games, they extend this framework to handle heterogeneous players with convex cost functions, which allows them to provide deeper insights into strategic interactions in public goods games. The paper is well-written for the most part. The authors present a fairly complete picture of the equilibria of these games in Section 3.

A minor criticism is that I am not sure what the takeaway message of the case study is. For example, I am not sure what the sigificance of Theorem 4.2 is, within the broader context of public goods games. I also would have liked to see some empirical validation of the results, even if it were just on simulated data.

**Questions:**

n/a

**Reviewer Confidence:**

3: The reviewer is confident but not certain that the evaluation is correct

**Scope:**

4: The work is relevant to the Web and to the track, and is of broad interest to the community

---

### Official Review · Reviewer_oCXC · 2024-11-27

**Novelty:** 5
**Technical Quality:** 5

**Review:**

**Summary.**
This paper studies the networked digital public goods problem and in particular, focuses on the setting with non-linear costs (unlike most prior works). They also assume that the utilities are heterogeneous. More specifically, each agent has a concave benefit function and a convex cost function both of which essentially model diminishing returns. The goal of the paper is to understand the existence and uniqueness of Nash equilibria in this model.

To prove the existence of NE, the paper utilizes Brouwer’s fixed point theorem. Then they show (Example 1) that the NE may not be unique. Thus, this provides some motivation for understanding when NEs are unique. For this, they define a notion of near-potential games. They then show that near-potential games have unique NEs which in turn implies that the original problem has unique NEs with some assumptions.

The authors then consider the sensitivity of the player’s utilities when the problem changes by small amounts. The authors also give some application of their results. In particular, they show that if the non-diagonal elements are i.i.d. then unique NEs exist. They also show that if the marginal gain matrix is upper-triangular then NEs also exist.

**Review.**
Overall, I think this is an interesting paper that studies networked digital public goods games and proves some nice theoretical results (particularly, the existence and uniqueness of NEs). The techniques seem interesting and I think would appeal to researchers that study NEs. The topic of the paper is certainly a fit for WWW and I’m sure some researchers would be interested in engaging with the authors.

The main downside of the paper is that I think it would be nice to see additional results. For example, I would be interested in the price of anarchy of such equilibria (irrespective of whether NEs are unique or not). Have these been studied in the past? The authors also mention that in the homogeneous and linear setting, NEs have some nice structure (in particular, they are always maximal independent sets). It would also be interesting to see if there are structural results in this setting as well even in the homogenous setting but with non-linear utilities and costs.

**Comments.**
- Line 352: Can you explain why it is without loss of generality to assume $w_{ii} = 1$ for all $i$ without loss of generality? It was not clear to me when I was reading this. For example, let’s say that everybody except player 1 has a constant utility of 0. Then setting $w_{ii} = 1$ for all $i$ does not really make sense. But probably I am missing something so some explanation would be great. Is it because the paper is mainly only interested in the existence and uniqueness of NE?

**Minor comments.**
- Line 180: studies → study
- Line 194: cut will
- Line 343: publicly
- Line 398: ascend → ascent

**Questions:**

I wonder if the authors could comment on how the exponential rate in Lemma 3.2 is used in the proofs. I didn’t see much discussion about this.

**Reviewer Confidence:**

3: The reviewer is confident but not certain that the evaluation is correct

**Scope:**

4: The work is relevant to the Web and to the track, and is of broad interest to the community

---

### Official Review · Reviewer_p63g · 2024-12-01

**Novelty:** 5
**Technical Quality:** 5

**Review:**

This paper studies public good games while each player has a concave utility function and a convex cost function. It shows the existence of Nash Equilibrium (not necessarily unique) in their setting. It further provides an analysis of under which conditions NE is unique.

**Strengths**

- This paper is the first one that proposes and studies the public good games with convex cost, generalizing the previous results with linear cost functions. It also provides an almost full picture of the NE of the public good games with convex cost, including the existence of NE and the uniqueness of NE for some special cases.

- Overall, this paper is well-written and easy to follow. In the in-depth analysis of the uniqueness of NE, this paper not only provides three different conditions in which a unique NE exists but also provides intuitions on the conditions, providing a nice explanation behind the conditions.

**Weakness**

- While this paper provides a nice and detailed motivation for the convex cost, the motivation for the concave utility is not that clear. It seems to be well-designed to make the best-response function continuous, which leads to the existence of NE due to the fixed point theorem.

**Questions:**

**Question:**

Can you provide more motivation or intuition about the concave utility?

**Minor Comments**

- Use the correct CCS concepts.
- Line 477, an (pure) NE -> a (pure) NE.
- Line 482, an (pure strategy) NE -> a (pure strategy) NE.

**Reviewer Confidence:**

3: The reviewer is confident but not certain that the evaluation is correct

**Scope:**

4: The work is relevant to the Web and to the track, and is of broad interest to the community

---

### Official Review · Reviewer_Coy4 · 2024-12-02

**Novelty:** 6
**Technical Quality:** 6

**Review:**

# Paper summary

This paper studies a multi-player game for public goods under a network structure. In this model, there is a network where each node denotes a strategic player, whose action space is her effort to be invested. Each directed weighted edge $(i, j)$ specifies the linear impact of player $j$'s effort on player $i$. Each player's utility is her valuation (as a function of her own effort level and all other players' effort levels scaled by the edge weights) minus her cost (as a function of her effort level). The paper assumes that the valuation function is concave, and strictly increasing, and the cost function is convex and strictly increasing.

The results of this paper are as follows:

1. The authors design the best response dynamic and prove its convergence to the social-welfare optimal outcome [Theorem 2.2]. Exponential and linear convergence rates have been established with and without additional assumptions.

2. The authors prove the existence of the pure Nash equilibrium [Theorem 3.1].

3. The authors construct an instance [Example 1] to show that the pure Nash equilibrium may not be unique. Then, the authors identify a batch of conditions to ensure the uniqueness of pure Nash equilibrium [Theorems 3.3 -3.5].

4. The authors explore the game equivalence, which broadens the class of games with unique Nash equilibrium [Corollary 3.7].

5. Finally, the authors conduct case studies for their model in Section 4.

# Evaluation

- This paper contributes to the literature on networked public goods games. Studying quasi-linear utility and decreasing valuation margin is motivated. Unlike many prior works in this area, this paper does not impose strong assumptions (e.g., linear cost function, positive edge weights) in their model.

- The paper provides a comprehensive package, which includes the computation method for the optimal outcome, the existence and uniqueness of the pure Nash equilibrium, and the case studies.

- The results are solid and technically interesting. I enjoy reading this paper.

**Questions:**

It would be helpful if the authors could add some (theoretical/numerical) discussion about the price of anarchy (PoA) and price of stability (PoS).

**Reviewer Confidence:**

3: The reviewer is confident but not certain that the evaluation is correct

**Scope:**

3: The work is somewhat relevant to the Web and to the track, and is of narrow interest to a sub-community

---

### Official Review · Reviewer_aWH4 · 2024-12-02

**Novelty:** 5
**Technical Quality:** 5

**Review:**

This paper studies the problem of networked public goods games with heterogeneous players, concave values, and convex costs. The authors first establish the existence of Nash Equilibrium (NE) in this model and provide conditions under which NE is unique. The authors further investigated how slight modifications in the model via money redistribution may impact players' utilities and applied the theoretical results to an example inspired by Fershtman and Nitzan [1991].

Strength: The collection of the theoretical results is non-trivial and technically strong.

Weakness: The paper would be stronger if more motivations/justifications for the conditions in Section 3.2. can be provided.

**Questions:**

1. In the abstract, the authors claim that the results have "significant implications for policy design in internet economic and social networks". Could the authors elaborate a bit more on which results have such implications? In particular, most results in this paper are centered around existence and uniqueness of NE and it is not immediate to the reviewer how these results have implications for policy design.

2. Could the authors provide further motivations/justifications for the conditions in Section 3.2? For example, are there real-world scenarios that can be modeled as a game with those conditions?

3. Is it possible to study the performance of equilibrium (for example, the price of anarchy for the games being studied)?

**Reviewer Confidence:**

2: The reviewer is willing to defend the evaluation, but it is likely that the reviewer did not understand parts of the paper

**Scope:**

3: The work is somewhat relevant to the Web and to the track, and is of narrow interest to a sub-community